# ONLINE CONTINUAL LEARNING VIA PURSUING CLASS-CONDITIONAL FUNTION

## ABSTRACT

Online continual learning is a challenging problem where models must learn from a non-stationary data stream while avoiding catastrophic forgetting. Inter-class imbalance during training has been identified as a major cause of forgetting, leading to model prediction bias towards recently learned classes. In this paper, we theoretically analyze that inter-class imbalance is entirely attributed to imbalanced class-priors, and the class-conditional function learned from intra-class distributions is the Bayes-optimal classifier. Accordingly, we present that a simple adjustment of model logits during training can effectively resist prior class bias and grasp the corresponding Bayes-optimum. Our method mitigates the impact of inter-class imbalance not only in class-incremental but also in realistic general setups by eliminating class-priors and pursuing class-conditionals, with minimal additional computational cost. We thoroughly evaluate our approach on various benchmarks and demonstrate significant performance improvements compared to prior arts. For example, our approach improves the best baseline by 4.6% on CIFAR10.[1]

## 1 INTRODUCTION

Continual learning (CL) has emerged to equip deep learning models with the ability to handle multiple tasks on an unbounded data stream. This paper focuses on the online class-incremental learning (class-IL) problem (Zhou et al., 2023), which holds high relevance to real-world applications (Zhou et al., 2023). In online CL, also known as task-free CL, data is obtained from an unknown non-stationary stream for single-pass training. Class-IL, in contrast to task-incremental learning (task-IL), continuously introduces new classes to the model as the data stream distribution evolves, without task-identifiers to assist classification.

*Catastrophic forgetting* (Goodfellow et al., 2014) is a major obstacle to deploying deep learning models in CL. Recent research attributes catastrophic forgetting to *recency bias* (Chrysakis & Moens, 2023), which enforces deep neural networks to classify samples into currently learned classes. Previous works (Wang et al., 2023) have observed that one of the primary causes of recency bias is *inter-class imbalance* throughout training, which is more severe in online CL than offline. Growing attention to inter-class imbalance has given rise to online CL methods to alleviate the negative impact of imbalance, among which recently rehearsal-based methods have been highly successful. Some methods (Guo et al., 2022) train only on memory samples, and some (Caccia et al., 2022) separate gradients of incoming and replayed classes to obtain balanced learning, while they fail to fundamentally address inter-class imbalance in online CL, particularly on large-scale data streams (Wang et al., 2023).

Upon decomposing sample probability in non-stationary data streams through conditional probability (sample probability = class-prior × class-conditional), we reveal that recency bias caused by inter-class imbalance is entirely attributable to imbalanced class-priors and independent of class-conditionals. The underlying *class-conditional invariant* in online class-IL data streams motivates us to learn a function from intrinsic intra-class distributions instead of traditional sample distributions. We theoretically achieve that the class-conditional function is the Bayes-optimal classifier that minimizes the class-balanced error. In light of the above considerations, we propose handling class-priors and class-conditionals separately to help address inter-class imbalance and learn a balanced

---

[1]The code of implementation is available in Supplementary Material.

classifier in online CL. Beyond online class-IL, our proposal can also facilitate the realistic online general continual learning (GCL) (Xu et al., 2021) where both class-priors and class-conditionals are time-varying. We show that preserving knowledge in the class-conditional function can better adapt the learner to changing domains.

Our attention to the class-conditional function leads to an effective and efficient logit adjustment method that adjusts model-predicted logits via input class-priors during training to pursue a balanced classifier. Our approach provides the following three practical benefits in compared other previous online CL methods: (1) It can eliminate the prediction bias caused by the imbalance between old and new classes, as well as the inherent inter-class imbalance of the data stream. (2) It is orthogonal to the methods improving replay strategies and plug-in to most of the rehearsal-based methods. (3) It improves performance with nearly no additional computational overhead.

We conduct comprehensive experiments over various datasets. Our method lifts the plainest Experience Replay (ER) (Chaudhry et al., 2019) to state-of-the-art performance in online class-IL and GCL setup, e.g., improving the accuracy of the best baseline by 4.6% on CIFAR10 (Krizhevsky, 2009). Furthermore, we notice that inter-class imbalance dominates forgetting in long sequential data streams, which is rarely evaluated and always underestimated in previous work, so we evaluate on the challenging ImageNet (Deng et al., 2009) and iNaturalist (Horn et al., 2017), where our proposed method consistently outperforms previous approaches.

Key contributions of this paper include: (1) We discover the class-conditional invariant and the Bayesian optimality of the class-conditional function in online class-IL. (2) We propose eliminating class-priors and learning class-conditionals separately under general online setup. (3) We introduce to adjust model logit outputs in training with a batch-wise sliding-window estimator for time-varying class-priors to pursue the class-conditional function.

## 2 PROBLEM SETUP

Beyond the task-IL setting (Li & Hoiem, 2016) with clear task-boundaries, we consider a more realistic environment where task-identifiers and task-boundaries are absent at any time, and the total number of labels is unknown. Specifically, let $\mathcal{X}$ be the instance set and $\mathcal{Y}$ be the corresponding label set. In online CL, $|\mathcal{Y}| = \infty$. At time $t \in \mathcal{T} = \{1, 2, \dots\}$, given an unknown non-stationary data stream $\mathcal{D}_t$ over $\mathcal{X} \times \mathcal{Y}$, the learner samples data batch $B_t = \{x_i, y_i\}_{i=1}^{|B_t|} \sim \mathbb{P}(x, y|\mathcal{D}_t)$. We refer to $B_t$ as the *incoming batch*. If a pair of instance and label is not stored in the memory, it will be inaccessible in subsequent training unless resampled.

Commonly, a constrained memory $\mathcal{M}$ ($|\mathcal{M}| \leq M$) is utilized to enhance online CL: if the buffer is not empty at time $t$, a *Retrieval* program ensembles several instances and other specific information $I$ to form a *buffer batch* $B_t^{\mathcal{M}} = Retrieval(B_t, \mathcal{M}_t) = \{x_i, I_i\}_{i=1}^{|B_t^{\mathcal{M}}|} \sim \mathbb{P}(x, I|\mathcal{M}_t)$. The buffer *Update*s with incoming batches, $\mathcal{M}_{t+1} \leftarrow Update(B_t, \mathcal{M}_t)$. Typically, ER (Chaudhry et al., 2019) stores instances and labels $I_i = y_i$, retrievals by random replaying, and updates via reservoir sampling (Vitter, 1985). Rehearsal helps to alleviate inter-class imbalance when the number of classes is limited, but can not fundamentally eliminate its impact. The minimum class-prior in memory is bounded by the inverse proportion to the number of observed classes, $\min_{y \in \mathcal{Y}_t} \mathbb{P}(y|\mathcal{M}_t) \leqslant 1/|\mathcal{Y}_t| \to 0$ ($t \to \infty$). When the number of seen classes surges, rehearsal will no longer be able to support balanced inter-class learning.

The learner is a neural network parameterized by $\Theta = \{\theta, w\}$. Function $f_\theta : \mathcal{X} \to \mathbb{R}^D$ extracts feature embeddings with dimension $D$. Following the feature extractor, a single-head linear classifier produces logits, $\Phi(\cdot) = w^\top f_\theta(\cdot) : \mathcal{X} \to \mathbb{R}^{|\mathcal{Y}_t|}$ (for short $\Phi_y(\cdot) = w_y^\top f_\theta(\cdot)$), where $w \in \mathbb{R}^D \times \mathbb{R}^{|\mathcal{Y}_t|}$ represents the weights corresponding to target classes. The dimension of weights in the classifier can grow as more classes have been observed. The learner trains through a surrogate loss averaged on all input instances, $\mathcal{L}_t : \mathcal{Y}_t \times \mathbb{R}^{|\mathcal{Y}_t|} \to \mathbb{R}$ ($\mathcal{Y}_t$ is the set of all observed labels), typically the softmax cross-entropy loss:

$$\mathcal{L}_{\text{CE}}(y, \Phi(x)) = -\log \frac{e^{\Phi_y(x)}}{\sum_{y' \in \mathcal{Y}_t} e^{\Phi_{y'}(x)}} = \log[1 + \sum_{y' \neq y} e^{\Phi_{y'}(x) - \Phi_y(x)}]. \tag{1}$$

## 3 STATISTICAL VIEW FOR TIME-VARYING DISTRIBUTION LEARNING

The standard CL methods learn from the sample probability $\mathbb{P}(x, y|\rho_t)$ of the target distribution $\rho_t$ (for example, $\mathcal{D}_t$ in practice). The model is encouraged to pursue a posterior function $\propto \mathbb{P}(y|x, \rho_t)$ and to minimize the misclassification error $\mathbb{E}_{\rho_t}[\mathbb{E}_{x, y|\rho_t}[y \neq \arg\max_{y' \in \mathcal{Y}_t} \Phi_{y'}(x)]]$. From Bayesian and conditional probability rule, we notice $\mathbb{P}(y|x, \rho_t) \propto \mathbb{P}(x, y|\rho_t) = \mathbb{P}(x|y, \rho_t) \cdot \mathbb{P}(y|\rho_t)$, revealing that the sample probability $\mathbb{P}(x, y|\rho_t)$ of a time-varying distribution is controlled by the class-conditional $\mathbb{P}(x|y, \rho_t)$ and the class-prior $\mathbb{P}(y|\rho_t)$. *In unknown non-stationary data streams, the inter-class imbalance is entirely attributed to time-varying class-priors and is independent of class-conditionals.* Therefore, such a factorization of probability motivates us to learn a class-balanced classifier by exclusively pursuing a **class-conditional function** $\propto \mathbb{P}(x|y, \rho_t)$, which is agnostic to arbitrarily imbalanced class-priors. The class-conditional function has been widely studied in statistical learning on stable distributions (Collell et al., 2016), while we first introduce it to non-stationary stream distribution learning. In fact, we discover the class-balanced Bayes-optimality of the class-conditional function when learning stream distributions without domain drift, i.e., with fixed class-conditionals, as demonstrated in the following Theorem 3.1.

**Theorem 3.1.** *For the time-varying distribution $\rho_t$, given that its class-conditionals keep the same throughout time, i.e., $\forall t, \mathbb{P}(x|y, \rho_t) = \mathbb{P}(x|y, \rho_0)$, the class-conditional function satisfies the Bayes-optimal classifier $\Phi_t^*$ that minimizes the class-balanced error,*

$$\Phi_t^* \in \arg\min_{\Phi: \mathcal{X} \to \mathbb{R}^{|\mathcal{Y}_t|}} \text{CBE}(\Phi, \mathcal{Y}_t), \quad \arg\max_{y \in |\mathcal{Y}_t|} \Phi_{t, y}^*(x) = \arg\max_{y \in |\mathcal{Y}_t|} \mathbb{P}(x|y, \rho_t). \quad (2)$$

$$\text{CBE}(\Phi, \mathcal{Y}_t) = \frac{1}{|\mathcal{Y}_t|} \sum_{y \in \mathcal{Y}_t} \mathbb{E}_{\rho_t}[\mathbb{E}_{x|y, \rho_t}[y \neq \arg\max_{y' \in \mathcal{Y}_t} \Phi_{y'}(x)]]. \quad (3)$$

In other words, the Bayes-optimal class-balanced estimate is the class under which the sample is most likely to appear. $\text{CBE}(\Phi, \mathcal{Y}_t)$ is the Class-Balanced Error Menon et al. (2013) on the current label set $\mathcal{Y}_t$, extended from the misclassification error for class-balanced evaluation, formally in Equation 3. While bias towards recently occurring classes does not contribute to reducing the class-balanced error, approximation towards the true underlying class-conditionals aids in achieving balanced classification. This is because the class-balanced error is computed by averaging the per-class error rates. **Therefore, to address the impact of inter-class imbalance and leverage knowledge from intra-class intrinsic distributions, we propose eliminating class-priors and constructing a class-conditional function in online CL.** The proof of Theorem 3.1 is in Appendix A. Following, we discuss two distinct CL scenarios on the critical condition of class-conditionals.

**Discussion on online class-IL with time-invariant class-conditionals.** Prior works (Chrysakis & Moens, 2023) have typically assumed no occurrence of domain drift during the learning process in online class-IL. Although domain drift should be considered in realistic scenarios, nearly time-invariant class-conditionals are genuinely feasible in practical situations. For instance, acting as a lifelong species observer in the wild, the agent can find that the target class-conditionals conform to their natural distributions, determined by population semantic information and occurrence frequencies. Without intentional human interference, natural semantics will remain almost unchanged over a prolonged time, i.e., $\forall t, \mathbb{P}(x|y, \rho_t) \approx \mathbb{P}(x|y, \rho_0)$. In the experiments, we mainly adhere to the conventional class-IL configuration of no consideration of domain drift and focus on addressing the issues of inter-class imbalance and forgetting induced by recency bias.

**Discussion on online GCL with time-varying class-conditionals.** Online GCL (class- and domain-IL) is widely recognized as one of the most challenging real-world scenarios. In GCL both inter-class imbalance and intra-class domain drift are crucial considerations, since $\mathbb{P}(y|\rho_t)$ and $\mathbb{P}(x|y, \rho_t)$ fluctuate as data stream flows. While GCL has been studied in offline incremental setups (Xie et al., 2022), there has been no research on this topic under online conditions, to the best of our knowledge. Our proposal based on class-conditional function can be applied to online GCL to bridge this gap. As domain distributions vary in data stream, the class-conditional function should not favor any specific domain but blend all observed domains uniformly for optimal decision-making,

$$\arg\max_{y \in |\mathcal{Y}_t|} \Phi_{t, y}^*(x) = \arg\max_{y \in |\mathcal{Y}_t|} \frac{1}{t} \sum_{i=1}^{t} \mathbb{P}(x|y, \rho_i). \quad (4)$$

Since previous distributions are unavailable in CL, determining the optimal uniform domain distribution is intractable. Nevertheless, the disparity between the Bayes-optimal classifier and the learned class-conditional function could be measured by the similarity between their underlying intra-class distributions. We combine with knowledge distillation techniques (Li & Hoiem, 2016; Tao et al., 2020) to narrow that disparity in probability space. Results in §6.2 show that preserving the knowledge in the class-balanced class-conditional function after eliminating class-priors can better adapt to domain drift than the standard posterior function. Therefore, our proposal is scalable to realistic general online settings. Furthermore, it paves the way for developing further efficient solutions for online GCL by minimizing the intra-class probabilities gap from the optimal class-conditional function, which we intend to explore in future research.

# 4 METHOD

## 4.1 LOGIT ADJUSTMENT TECHNIQUE

Now our objective becomes excluding class-priors and establishing a function for current class-conditionals, i.e., $\Phi_t : \mathcal{X} \to \mathbb{R}^{|\mathcal{Y}_t|}$, $\exp(\Phi_{t,y}) \propto \mathbb{P}(x|y, \rho_t)$. However, it is notoriously difficult to model the class-conditionals explicitly. To detour this problem, we draw on the Logit Adjustment technique proposed by Menon et al. (2021). Suppose the optimum scorer obtained by minimizing misclassification error on the target distribution $\rho_t$ at time $t$ is $s_t^* : \mathcal{X} \to \mathbb{R}^{|\mathcal{Y}_t|}$, $\exp(s_{t,y}^*) \propto \mathbb{P}(y|x, \rho_t)$. Recalling $\mathbb{P}(y|x, \rho_t) \propto \mathbb{P}(x|y, \rho_t) \cdot \mathbb{P}(y|\rho_t)$, we can derive the relationship between the class-conditional function $\Phi_t$ and the optimum scorer $s_t^*$ as follows:

$$\underset{y \in |\mathcal{Y}_t|}{\arg\max}\, e^{s_{t,y}^*(x)} = \underset{y \in |\mathcal{Y}_t|}{\arg\max} \left( e^{\Phi_{t,y}(x)} \cdot \mathbb{P}(y|\rho_t) \right) = \underset{y \in |\mathcal{Y}_t|}{\arg\max} \left( \Phi_{t,y}(x) + \ln \mathbb{P}(y|\rho_t) \right). \quad (5)$$

Equation 5 induces a straightforward method to approximate class-conditionals and to achieve a class-balanced classifier: adjusting the model logits output according to class-priors $\mathbb{P}(y|\rho_t)$ and directly optimizing the softmax cross-entropy loss.

## 4.2 LOGIT ADJUSTED SOFTMAX CROSS-ENTROPY LOSS

We now show how to incorporate Logit Adjustment technique into the softmax cross-entropy loss in online CL. We aim to pursue the class-conditional function and to address the inter-class imbalance issues in online CL. The modified Logit Adjusted Softmax cross-entropy loss is defined as follows:

$$\mathcal{L}_{\text{LAS}}(y, \Phi(x)) = -\log \frac{e^{\Phi_y(x) + \tau \cdot \log \pi_{y,t}}}{\sum_{y' \in \mathcal{Y}_t} e^{\Phi_{y'}(x) + \tau \cdot \log \pi_{y',t}}} = \log[1 + \sum_{y' \neq y} \left( \frac{\pi_{y',t}}{\pi_{y,t}} \right)^\tau \cdot e^{\left( \Phi_{y'}(x) - \Phi_y(x) \right)}], \quad (6)$$

where $\tau$ is the temperature scalar, and $\pi_{y,t}$ is the class prior $\mathbb{P}(y|\mathcal{S}_t)$ at time $t$. In practice, $\mathcal{S}_t$ represents the data point collection from which the model samples input batch each time. Due to the uncertainty of $\mathcal{S}_t$, it is impossible to pinpoint class priors at each moment. To overcome this barrier, the following §4.3 will provide a simple yet effective method for estimating class-priors in the flowing input stream. Applied to rehearsal-based methods, $\mathcal{L}_{\text{LAS}}$ will act both on incoming and buffer batches to fully exploit input data. The right-hand side of Equation 6 illustrates its distinction to the cross-entropy loss in Equation 1, enforcing a large relative margin between the major class and the minor class, i.e., $(\pi_{\text{major},t}/\pi_{\text{minor},t})^\tau > 1$.

## 4.3 ESTIMATOR FOR TIME-VARYING CLASS-PRIORS

When facing an unknown time-varying input data stream, it is required to continuously estimate class-priors $\pi_{y,t}$ for $\mathcal{L}_{\text{LAS}}$ at each time $t$, instead of simply determining class-priors based on a large amount of training data as in stationary distribution (Collell et al., 2016). Therefore, we propose an intuitive batch-wise estimator with a sliding window, where *the occurrence frequency of a label in input batches covered by the sliding time window approximates the corresponding class-prior to that label*. Given the length $l > 0$ of the time frame, $\pi_{y,t}$ is calculated as follows:

$$\pi_{y,t} = \frac{\sum_{i=t-l+1}^{t} \sum_{\{x',y'\} \in B_i^{\mathcal{S}}} \mathbb{1}(y' = y)}{\sum_{i=t-l+1}^{t} |B_i^{\mathcal{S}}|}, \quad (7)$$

Figure 1: Left is the diagram of Experience Replay (ER) with our proposed Logit Adjusted Softmax and a batch-wise sliding-window estimator (ER-LAS). LAS helps mitigate the inter-class imbalance problem by adding label frequencies to predicted logits. The model in ER-LAS is still trained via the softmax cross-entropy loss. And right is model prediction test samples by Fine-Tune, ER, and ER-LAS on C-CIFAR100 (10 tasks). The gray dashed line indicates the ground truth task-wise distribution ($1k$ for each). We count according to the tasks to which the predicted classes belong.

where $\mathbb{1}(\cdot)$ is the indicator function of label $y$ and $B_t^{\mathcal{S}}$ is the input batch sampled from the data point collection $\mathcal{S}_t$. For rehearsal-based methods, the input batch often consists of the incoming and buffer batch, i.e., $B_t^{\mathcal{S}} = B_t \cup B_t^{\mathcal{M}}$. The length $l$ of the time window concerns a sensitivity-stability trade-off (Nagengast et al., 2011) with respect to the estimation of class priors, which we further study in the sensitivity analysis of §6.3.

**Discussion.** Logit Adjusted Softmax cross-entropy loss and the batch-wise sliding-window estimator together constitute our proposed **LAS** approach to pursue the class-conditional function. Our method is orthogonal to previous methods of various replay strategies and knowledge distillation techniques. Exact joint label distribution of the non-stationary data stream and the memory retrieval program is unnecessary to our approach, allowing us to effortlessly incorporate LAS into existing methods and correct their model prediction bias caused by inter-class imbalance at nearly no cost of additional computational overhead. The experiment in Figure 1(right) verifies the effect of LAS on correcting the prediction bias, which follows the same setting as in §6. Fine-Tune, which trains without any precautions against catastrophic forgetting and inter-class imbalance, categorizes all test samples into the most recently studied task classes. ER (Chaudhry et al., 2019) includes a constrained memory to store previously observed data but still assigns about 38% (instead of the expected 10%) test samples to the most recently learned classes. By contrast, our ER-LAS shown in Figure 1(left) eliminates the recency bias, achieves balanced class-posteriors similar to ground truth distribution, and significantly improves ER performance evaluated in the following §6. The algorithm of LAS is in Appendix B.

**Implementation in online GCL.** We combine LAS with knowledge distillation in online GCL to preserve a class-balanced class-conditional function over averaged domain distributions. We directly calculate the distillation loss between the outputs of old and current models without logit adjustment. Noting that distillation necessitates well-defined task-boundaries to preserve the previous model for distillation. This requirement presents a formidable obstacle in online CL settings, where such boundaries are absent. To investigate the efficacy of our proposed method under the online GCL setting, we allow to acquire task-boundaries in relative experiments. The algorithm of LAS with knowledge distillation for online GCL is in Appendix B.

## 5    RELATED WORK

We next provide some intuition on the effectiveness of our proposed approach by comparing LAS to prior work from the perspective of traditional and continual imbalanced distribution learning. We also highlight the computational efficiency in online conditions.

**Methods for mitigating inter-class imbalance in stable distributions. Logit Adjustment** (Menon et al., 2021) technique appears similar to **Loss weighting** (Cui et al., 2019) methods, yet the two differ significantly in addressing inter-class imbalance. While Loss weighting methods can balance the representation learning on minority class samples by weighting after the loss between logits and ground truth, it cannot rectify prior class bias and therefore cannot address recency bias. In contrast, Logit Adjustment technique directly balances the class-priors on logits, eradicating the impact of prior class imbalance on model classification and resolving recency bias. In addition to Loss weighting methods, there are also other methods such as **Weight normalization** (Kang et al., 2020), **Resampling** (Kubát & Matwin, 1997), and **Post-hoc correction** (Collell et al., 2016). Different

from these methods and the original Logit Adjustment technique, our adapted LAS possesses firm statistical grounding for non-stationary distributions. We compare with these inter-class imbalance mitigation methods in Appendix F.2.

**Methods for mitigating inter-class imbalance in non-stationary distributions.** The fundamental **ER** (Chaudhry et al., 2019) and recently proposed **ER-ACE** (Caccia et al., 2022) represent two extreme cases of our approach. ER corresponds to the case where $\tau = 0$, and $\mathcal{L}_{\text{LAS}}$ degenerates into the conventional cross-entropy loss function $\mathcal{L}_{\text{CE}}$ in Equation 1, losing the ability to alleviate inter-class imbalance. ER-ACE employs asymmetric losses for incoming and buffer batches, considering only the classes present in the current batch for incoming, i.e., $\tau \to \infty$, and all previously seen classes for replaying, i.e., $\tau = 0$, to mitigate representation shift. However, completely separating the gradients of current and past classes blocks the construction of inter-class decision boundaries. Our method lies between ER and ER-ACE, not only pursuing class-conditional function but also encouraging large relative margins between old and new classes in online class-IL, i.e., always $(\pi_{\text{new},t}/\pi_{\text{old},t})^\tau \gg 1$ in Equation 6 derived from their imbalance. We also notice highly related **Logit Rectify** methods (Zhou et al., 2023) designed for offline task-IL, which we compare in Appendix F.3.

**Computational efficiency.** Online CL cannot ignore real-time requirements because memory and training time is usually limited in practical scenarios. Compared to traditional Softmax, Logit Adjusted Softmax slightly increases the computational cost of $\mathcal{O}(|\mathcal{Y}_t|)$. Our suggested estimator raises the calculation time by $\mathcal{O}(|B_t| + |B_t^{\mathcal{M}}| + |\mathcal{Y}_t|)$ and the memory cost by $\mathcal{O}(|\mathcal{Y}_t|)$. In contrast to the time and storage overhead of the model and the memory, such an increase is negligible and lower than in previous works. Our experiments primarily compare methods with computational costs similar to our approach. Noting that CL methods based on contrastive learning (Guo et al., 2022) may consume substantially more computational resources than our algorithm. We present a performance comparison with these methods in Appendix F.4.

# 6 EXPERIMENT

In this section, we conduct comprehensive experiments to demonstrate the effectiveness of pursuing the class-conditional function via our proposed LAS in online CL. First, we investigate the performance of LAS in the online class-IL scenario with class-disjoint tasks. Then, we evaluate LAS's gains on rehearsal-based methods in the online class-IL setup and gains on our proposal with knowledge distillation approaches in the online GCL setup. Finally, we study the extreme variants of our method, the necessity of the suggested batch-wise estimator with a sliding window, and the hyperparameter sensitivity of our LAS.

**Benchmark setups.** We use 5 image classification datasets combined with 2 kinds of CL setups to form 6 benchmarks. Among datasets, **CIFAR10** (Krizhevsky, 2009) has 10 classes. **CIFAR100** (Krizhevsky, 2009) has 100 classes, and they can also be categorized into 20 superclasses with 5 domains. **TinyImageNet** (Le & Yang, 2015) has 200 classes. **ImageNet** ILSVRC 2012 Deng et al. (2009) has 1,000 classes, evaluating method performance on the long sequence data stream. **iNaturalist** 2017 (Horn et al., 2017) has 5,089 classes. The distribution of images per category in iNaturalist follows the observation frequency of the species in the wild, so the data stream possesses inherent inter-class imbalance. As to CL setups, **online class-IL (C)** (Aljundi et al., 2019) splits a dataset into multiple tasks with uniform disjoint classes, e.g., C-CIFAR10 (5 tasks) is split into 5 disjoint tasks with 2 classes each, except for C-iNaturalist (26 tasks) that is organized into 26 disjoint tasks according to the initial letter of each class. **Online GCL (G)** Xie et al. (2022) covers class- and domain-IL setup, where incoming data contains images from new classes and new domains. We only apply online GCL on CIFAR100. The learner needs to predict superclass labels. Each superclass has 5 subclasses representing 5 different domains within the same class. We evaluate the final average accuracy $A_T$ and the final average forgetting $F_T$ (Chaudhry et al., 2020). See Appendix C and Appendix D for more details about benchmark setups and metrics.

**Training Protocol.** For all experiments, unless otherwise specified, following Buzzega et al. (2020). We use the full ResNet18 as the feature extractor. For small-scale datasets, we start training from scratch. We pre-train models on 100 randomly selected classes from C-ImageNet and then perform online learning on the remaining 900 classes(Gallardo et al., 2021). As for C-iNaturalist, we pre-train models on the entire ImageNet dataset. A single-head classifier is applied to classify all seen labels.

Table 1: Final average accuracy $A_T$ (higher is better) on C-CIFAR10 (5 tasks), C-CIFAR100 (10 tasks), and C-TinyImageNet (10 tasks). $M$ is memory size.

| Dataset | C-CIFAR10 | | | C-CIFAR100 | | | C-TinyImageNet | | |
|---|---|---|---|---|---|---|---|---|---|
| Method | $M = 0.5k$ | $M = 1k$ | $M = 2k$ | $M = 0.5k$ | $M = 1k$ | $M = 2k$ | $M = 0.5k$ | $M = 1k$ | $M = 2k$ |
| ER | $40.9_{\pm1.2}$ | $45.4_{\pm1.8}$ | $50.3_{\pm1.1}$ | $12.9_{\pm0.3}$ | $16.5_{\pm0.4}$ | $19.8_{\pm0.6}$ | $8.8_{\pm0.2}$ | $11.0_{\pm0.2}$ | $14.3_{\pm0.3}$ |
| DER++ | $49.4_{\pm1.0}$ | $49.7_{\pm3.0}$ | $48.9_{\pm0.9}$ | $8.9_{\pm0.4}$ | $13.1_{\pm0.4}$ | $12.3_{\pm0.4}$ | $5.9_{\pm0.2}$ | $8.0_{\pm0.3}$ | $9.5_{\pm0.3}$ |
| MRO | $43.4_{\pm1.0}$ | $49.3_{\pm1.1}$ | $55.9_{\pm0.6}$ | $11.5_{\pm0.1}$ | $18.3_{\pm0.2}$ | $23.1_{\pm0.1}$ | $5.9_{\pm0.1}$ | $9.2_{\pm0.1}$ | $13.4_{\pm0.2}$ |
| SS-IL | $47.7_{\pm0.7}$ | $52.6_{\pm0.5}$ | $51.7_{\pm0.4}$ | $19.2_{\pm0.2}$ | $21.5_{\pm0.2}$ | $24.2_{\pm0.2}$ | $13.1_{\pm0.2}$ | $14.9_{\pm0.1}$ | $17.1_{\pm0.9}$ |
| CLIB | $48.4_{\pm0.9}$ | $54.8_{\pm1.0}$ | $55.9_{\pm1.0}$ | $15.9_{\pm0.2}$ | $20.7_{\pm0.2}$ | $25.3_{\pm0.3}$ | $8.3_{\pm0.1}$ | $12.1_{\pm0.2}$ | $15.9_{\pm0.2}$ |
| ER-ACE | $44.4_{\pm1.0}$ | $48.1_{\pm1.1}$ | $51.2_{\pm1.2}$ | $18.6_{\pm0.4}$ | $22.5_{\pm0.5}$ | $25.0_{\pm0.9}$ | $11.4_{\pm0.2}$ | $14.8_{\pm0.2}$ | $16.4_{\pm0.4}$ |
| ER-OBC | $45.1_{\pm0.6}$ | $46.4_{\pm0.6}$ | $46.0_{\pm0.4}$ | $15.6_{\pm0.2}$ | $17.9_{\pm0.2}$ | $22.1_{\pm0.3}$ | $9.1_{\pm0.1}$ | $13.2_{\pm0.1}$ | $16.4_{\pm0.1}$ |
| ER-LAS | $\mathbf{51.7_{\pm0.9}}$ | $\mathbf{55.3_{\pm1.6}}$ | $\mathbf{60.5_{\pm0.8}}$ | $\mathbf{20.1_{\pm0.2}}$ | $\mathbf{25.7_{\pm0.3}}$ | $\mathbf{27.0_{\pm0.3}}$ | $\mathbf{13.7_{\pm0.2}}$ | $\mathbf{15.5_{\pm0.2}}$ | $\mathbf{18.7_{\pm0.2}}$ |

We use SGD optimizer without momentum and weight decay. The learning rate is set to 0.03 and kept constant. Incoming and buffer batch sizes are both 32. On C-ImageNet and C-iNaturalist, we set both batch sizes to 128. We apply standard data augmentation, including *random-resized-crop*, *horizontal-flip*, and *normalization*. Some literature(Koh et al., 2022) assumes that data arrive one-by-one in online CL, in which case we can accumulate samples as a batch to help model optimization convergence. We discuss the performance under varying batch sizes and per-sample updating in Appendix F.1. For online CL, only one epoch is used to run all methods for each task, and gradient descent is performed only once per incoming batch. By default, we set $\tau = 1.0$ and $l = 1$ for LAS. We report means and standard deviations of all results across 10 independent runs.

**Baselines.** We consider 7 rehearsal-based methods for online CL to compare: **ER** (Chaudhry et al., 2019) uses reservoir update and random replay. **DER++** (Buzzega et al., 2020) replays samples with previous logits for distillation loss. **MRO** (Chrysakis & Moens, 2023) only trains from memory. **SS-IL** (Ahn et al., 2020) separates the loss for present and absent classes. **CLIB** (Koh et al., 2022) updates by sample-wise importance and only trains on replayed samples. **ER-ACE** (Caccia et al., 2022) employs the asymmetric loss to reduce representation shift. **ER-OBC** (Chrysakis & Moens, 2023) additionally updates the classifier by balanced buffer batches. In addition, we enhance 3 methods of replay strategy: **MIR** Aljundi et al. (2019) retrieves the memory samples most interfered with by the model updating. **ASER**$_\mu$ (Shim et al., 2020) calculates Shapley values of samples to update and retrieve. **OCS** (Yoon et al., 2022) selects a coreset with high affinity to replay. Also, knowledge distillation losses in 3 approaches are augmented by LAS: **LwF** Li & Hoiem (2016) distills on logits of previous classes. **LUCIR** Hou et al. (2019) distills on normalized features. **GeoDL** (Simon et al., 2021) also distills in the feature space but measures by the geodesic path.

## 6.1 RESULTS ON ONLINE CLASS-IL SCENARIOS

**Accuracy results.** Table 1 and Table 2 show the final average accuracy for C-CIFAR10, C-CIFAR100, C-TinyImageNet, C-ImageNet, and C-iNaturalist with various memory sizes. ER-LAS consistently outperforms all compared baselines, achieving 60.5% (+4.6%), 27.0% (+1.7%), and 18.7% (+1.6%) on C-CIFAR10, C-CIFAR100, C-TinyImageNet respectively compared to the best baselines. Compared to only considering replayed samples in MRO and CLIB or separating the gradients between old and new classes in SS-IL and ER-ACE, LAS optimizes for a class-balanced function for incoming and buffer batches and enforces large relative margins between

Table 2: Final average accuracy $A_T$ (higher is better) and final average forgetting $F_T$ (lower is better) on C-ImageNet (90 tasks) and C-iNaturalist (26 tasks). We show the results of top-3 methods. Memory sizes are $M = 20k$.

| Dataset | C-ImageNet | C-iNaturalist |
|---|---|---|
| Method | $A_T \uparrow / F_T \downarrow$ | $A_T \uparrow / F_T \downarrow$ |
| ER | $31.8_{\pm0.1} / 38.6_{\pm0.2}$ | $4.7_{\pm0.0} / 18.0_{\pm0.0}$ |
| ER-ACE | $33.4_{\pm0.2} / 11.3_{\pm0.1}$ | $5.7_{\pm0.0} / 1.1_{\pm0.0}$ |
| MRO | $35.8_{\pm0.1} / 10.2_{\pm0.2}$ | $5.0_{\pm0.0} / \mathbf{0.4_{\pm0.0}}$ |
| ER-LAS | $\mathbf{39.3_{\pm0.1}} / \mathbf{9.0_{\pm0.1}}$ | $\mathbf{8.1_{\pm0.0}} / 2.8_{\pm0.0}$ |

imbalanced classes, resulting in better performance. Considering that the challenging C-ImageNet and C-iNaturalist benchmarks possess substantially longer sequences of data stream than the above three benchmarks, where the recency bias problem caused by inter-class imbalance becomes severely critical, we also apply LAS to boost the performance of ER. We present the results of the top-3 baselines (MRO, ER-ACE, ER) on C-ImageNet and C-iNaturalist. ER-LAS can obtain 39.3% (+3.5%) on C-ImageNet and 8.1% (+2.4%) on C-iNaturalist compared to the best baselines. Our extensive evaluations demonstrate the superior performance of our LAS by effectively alleviating inter-class

Table 3: Final average accuracy $A_T$ (higher is better) by replay strategy methods w/o and w/ LAS on C-CIFAR100 (10 tasks). Gains are shown in parentheses. $M$ is memory size.

| Dataset | C-CIFAR100 | |
|---|---|---|
| **Method** | $M = 0.1k$ | $M = 0.5k$ |
| ER | $6.5_{\pm 0.2}$ | $12.9_{\pm 0.3}$ |
| ER-LAS | $10.7_{\pm 0.2}(4.2\uparrow)$ | $20.1_{\pm 0.2}(7.2\uparrow)$ |
| MIR | $6.6_{\pm 0.3}$ | $12.0_{\pm 0.3}$ |
| MIR-LAS | $11.8_{\pm 0.1}(5.2\uparrow)$ | $21.1_{\pm 0.2}(9.1\uparrow)$ |
| $\text{ASER}_\mu$ | $7.8_{\pm 0.2}$ | $13.8_{\pm 0.3}$ |
| $\text{ASER}_\mu$-LAS | $9.5_{\pm 0.4}(1.7\uparrow)$ | $18.0_{\pm 0.3}(4.2\uparrow)$ |
| OCS | $9.4_{\pm 0.1}$ | $16.2_{\pm 0.2}$ |
| OCS-LAS | $12.7_{\pm 0.2}(3.3\uparrow)$ | $21.0_{\pm 0.3}(4.8\uparrow)$ |

Table 4: Final average accuracy $A_T$ (higher is better) by knowledge distillation approaches w/o and w/ LAS on G-CIFAR100 (20 tasks). Gains are shown in parentheses. $M$ is memory size.

| Dataset | G-CIFAR100 | |
|---|---|---|
| **Method** | $M = 0.1k$ | $M = 0.5k$ |
| ER | $20.4_{\pm 0.2}$ | $27.3_{\pm 0.4}$ |
| ER-LAS | $24.1_{\pm 0.2}(3.7\uparrow)$ | $31.5_{\pm 0.5}(4.2\uparrow)$ |
| LwF | $23.9_{\pm 0.3}$ | $30.1_{\pm 0.3}$ |
| LwF-LAS | $26.0_{\pm 0.2}(2.1\uparrow)$ | $32.4_{\pm 0.1}(2.3\uparrow)$ |
| LUCIR | $20.1_{\pm 0.1}$ | $29.4_{\pm 0.3}$ |
| LUCIR-LAS | $25.0_{\pm 0.2}(4.9\uparrow)$ | $32.6_{\pm 0.3}(3.2\uparrow)$ |
| GeoDL | $20.6_{\pm 0.2}$ | $30.1_{\pm 0.2}$ |
| GeoDL-LAS | $25.2_{\pm 0.2}(4.6\uparrow)$ | $32.8_{\pm 0.2}(2.7\uparrow)$ |

imbalance in the online class-IL setup with nearly no additional computation cost (Table 6). ER-LAS is only slightly slower than ER, contributing to its real-world online applications.

**Forgetting rate.** We compare the final average forgetting of ER-LAS with top-3 performed baselines (MRO, ER-ACE, ER) on C-ImageNet and C-iNaturalist. As shown in Table 2, ER-LAS achieves the least forgetting rate on C-ImageNet and only forgets more than MRO and ER-ACE on C-iNaturalist. However, the lowest forgetting rate (e.g., 0.4% of MRO) does not necessarily guarantee the highest accuracy (8.1% of ER-LAS) because of the stability-plasticity dilemma (Kim & Han, 2023). In the following sensitivity analysis of § 6.3, we show that although a lower forgetting rate can be obtained by deliberately tuning hyperparameters in our LAS, a better stability-plasticity trade-off can be achieved by the optimal hyperparameters. It is worth noting that in long sequence benchmarks, compared to ER without considering inter-class imbalance, methods trying to address recency bias not only remarkably reduce forgetting rates but also bring about improvements in accuracy, underscoring the importance of inter-class imbalance as a top priority in lifelong class-IL. We provide prediction results on C-ImageNet to further support our efficacy of eliminating recency bias in Appendix F.6. We also evaluate the final average forgetting on C-CIFAR10, C-CIFAR100, and C-TinyImageNet in Appendix F.5.

## 6.2 GAINS ON ENHANCED METHODS

**Rehearsal-based methods on online class-IL scenarios.** We verify the performance boost of LAS by plugging it into ER, MIR, $\text{ASER}_\mu$, and OCS. These three baselines train via softmax cross-entropy loss with different replay strategies, which harmonize with our approach. Table 3 shows that LAS can significantly improve ER and its variants (+1.7%~+9.1%) in the online class-IL setup. Although these methods with various memory management strategies benefit from our LAS, the gains depend on the estimation of class-priors from retrieval, as a relatively smaller boost is observed on $\text{ASER}_\mu$ which has a sophisticated strategy to manage memory.

**Knowledge distillation methods on online GCL scenarios.** To further investigate the effectiveness of our proposal based on estimating class-conditionals for the most difficult and realistic online GCL setup, we combined LAS with knowledge distillation approaches. Table 4 summarizes the results. Knowledge distillation losses obtain higher accuracy by adapting intra-class domain drift than plain rehearsal. Augmented by our LAS, consistent gains (+2.1%~+4.9%) are observed by eliminating class-imbalanced prior bias and optimizing the class-conditional function. The results demonstrate the validity of handling class-conditionals and class-priors separately in non-stationary stream learning. It also showcases the performance improvement of eliminating imbalanced class-priors by LAS in the GCL setup. Noting that we allowed the knowledge distillation methods to preserve old models at boundaries, which is intractable in real-world online CL. In future studies, we will explore the efficient and task-free method for handling intra-class domain drift to further refine the solution to online GCL.

Table 5: Ablation study about two extreme situations of $\tau$ and about randomly assigned ($Random$) or macro statistical ($Macro$) class-priors on C-CIFAR100 (10 tasks). $M = 2k$.

| Method | $\tau = 0$ | $\tau = \infty$ | $Random$ | $Macro$ | LAS |
|---|---|---|---|---|---|
| $A_T \uparrow$ | $19.4_{\pm 0.4}$ | $22.7_{\pm 0.2}$ | $20.6_{\pm 0.2}$ | $22.1_{\pm 0.6}$ | $\mathbf{27.0_{\pm 0.3}}$ |
| $F_T \downarrow$ | $29.1_{\pm 0.4}$ | $\mathbf{2.7_{\pm 0.4}}$ | $23.5_{\pm 0.2}$ | $14.2_{\pm 0.8}$ | $10.7_{\pm 0.4}$ |

Table 6: Training time compared with top-3 fast methods on C-CIFAR100 (10 tasks) with one Nvidia Geforce GTX 2080 Ti. $M = 2k$.

| Method | ER | ER-ACE | MRO | ER-LAS |
|---|---|---|---|---|
| Training Time (s) | 77.4 ($\times 0.94$) | 84.7 ($\times 1.02$) | 99.0 ($\times 1.20$) | 82.6 ($\times 1.00$) |

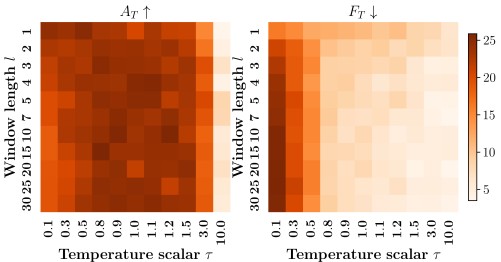

Figure 2: Final average accuracy (darker is better, left) and final average forgetting (lighter is better, right) of various hyperparameter combinations on C-CIFAR100 (10 tasks). $M = 2k$.

## 6.3 ABLATION STUDIES

**Extreme variants of LAS.** We investigate the performance of two variants of our method by pushing $\tau$ towards two extremes. When $\tau = 0$, LAS degenerates into the traditional softmax cross-entropy loss in ER. In $\tau = \infty$, we set $(\pi_{y',t}/\pi_{y,t})^\tau = 0$ in Equation 6 when $\pi_{y',t}/\pi_{y,t} < 1$, otherwise we keep this coefficient and set $\tau = 1$ to ensure runnable. It achieves a similar effect as separating the gradient of new and old categories in ER-ACE, reducing representation shift. As shown in Table 5, the performance of $\tau = 0$ is similar to ER as expected. $\tau = \infty$ benefits from a remarkably low forgetting rate. However, our proposed LAS with $\tau = 1$ achieves the highest accuracy, indicating that enforcing a relative margin between classes based on the imbalanced class-priors can obtain a better stability-plasticity trade-off.

**Necessity of batch-wise estimator with sliding window.** We empirically validate the necessity of our designed estimator. We randomly assign each prior of seen classes by a uniform distribution $U[0, 1]$ and normalize them to 1, as $Random$. We also explicitly calculate the joint label distribution of the current data stream and the memory replay, as $Macro$, which is intractable in practice. Results in Table 5 demonstrate that $Random$ degrades to performance similar to ER, and $Macro$ is also inferior to our proposed estimator. We conjecture that the online CL model is more concerned about the distribution within current or short-term input batches than the macro distribution of sequential data stream and memory. Therefore, our batch-wise estimator can help Logit Adjustment technique better exploit the class-conditional function to improve performance.

**Hyperparameter sensitivity analysis.** We conduct the sensitivity analysis of the hyperparameters $\tau$ and $l$ in our method in Figure 2. ER-LAS is robust to a wide range of $l$. In practice, if the distribution fluctuations in the stream can be discerned, we recommend setting short $l$ for streams that change rapidly and vice versa. As to temperature scalar $\tau$, it has distinct impacts on accuracy and forgetting rate. Although a larger $\tau$ can enable models to forget remarkably less, the best accuracy result is achieved around 1.0. Therefore, the stability-plasticity trade-off for target applications can be achieved by tuning $\tau$ and $l$ together.

## 7 CONCLUSION

We discover the class-conditional invariant and prove the Bayesian optimality of the class-conditional function in online class-IL. As a corollary of our theoretical analysis, we introduce Logit Adjusted Softmax with a batch-wise sliding-window estimator to purse the class-conditional function. Extended to online GCL, knowledge of the learned class-conditional function should be preserved for adaptation to domain drift. Extensive experiments demonstrate that LAS can achieve state-of-the-art performance on various benchmarks with minimal additional computational overhead, confirming the effectiveness and efficiency of our method to mitigate inter-class imbalance. It is effortless to implement LAS and plug it into rehearsal-based methods to correct their recency bias and boost their accuracy. Rehearsal-free approaches with LAS for online CL could be a subject of further study. Furthermore, we will continue to investigate efficient approaches to handling online domain drift, contributing to practical online GCL applications in the real world.

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

# A    PROOF

## A.1    PROOF IN ONLINE CLASS-INCREMENTAL SCENARIOS

To tackle the issue of inter-class imbalance, extensive research (Menon et al., 2013; Collell et al., 2016; Menon et al., 2021) has been conducted on the Bayes-optimal classifier for stable distributions. Actually, previous arts have proposed the following Theorem about the Bayes-optimal classifier:

**Theorem A.1.** *For time-invariant distributions, the Bayes-optimal estimate is the class under which the sample probability is most likely:*

$$\Phi^* \in \underset{\Phi:\mathcal{X}\to\mathbb{R}^{|\mathcal{Y}|}}{\arg\min} \ \mathrm{CBE}(\Phi, \mathcal{Y}), \quad \underset{y\in|\mathcal{Y}|}{\arg\max} \ \Phi_y^*(x) = \underset{y\in|\mathcal{Y}|}{\arg\max} \ \mathbb{P}(x|y) \tag{8}$$

Theorem A.1 Menon et al. (2013); Collell et al. (2016) states that the Bayes-optimal classifier is independent of arbitrary imbalanced label distributions $\mathbb{P}(y)$. The class-conditional function in stable distributions naturally minimizes the class-balanced error. From Theorem A.1 and given the condition of fixed class-conditionals, i.e., $\forall t, \mathbb{P}(x|y, \rho_t) = \mathbb{P}(x|y, \rho_0)$, we can derive the proof of Theorem 3.1 as follows:

$$\underset{y\in|\mathcal{Y}_t|}{\arg\max} \ \Phi_{t,y}^*(x) = \underset{y\in|\mathcal{Y}_t|}{\arg\max} \ \frac{1}{t}\sum_{i=1}^{t}\mathbb{P}(x|y, \rho_i) = \underset{y\in|\mathcal{Y}_t|}{\arg\max} \ \mathbb{P}(x|y, \rho_t) \tag{9}$$

## A.2    PROOF IN ONLINE GENERAL CONTINUAL LEARNING SCENARIOS

Without any prior information about the distribution of the test data, we assume that its distribution should conform to a uniformly joint distribution of all observed class distributions. Therefore, the final intra-class distribution is $\frac{1}{t}\sum_{i=1}^{t}\mathbb{P}(x|y, \rho_i)$. Therefore, the result of Equation 4 is from definition.

Let $p_{x|y}$ and $q_{x|y}$ be the underlying distributions the Bayes-optimal classifier and the learned class-conditional function represents, respectively. The class-balanced error gap between the Bayes-optimal classifier $\exp(\Phi_{t,y}^*(x)) \propto \mathbb{P}(x|y, p) = p_{x|y}$ and the learned class-conditional function $\exp(\Phi_{t,y}(x)) \propto \mathbb{P}(x|y, q) = q_{x|y}$ can be formalized as follows:

$$\underbrace{|\mathrm{CBE}(\Phi^*, \mathcal{Y}_t) - \mathrm{CBE}(\Phi, \mathcal{Y}_t)|}_{\epsilon_t(\Phi^*,\Phi)} \leqslant \underbrace{\frac{1}{|\mathcal{Y}_t|}\sum_{y\in\mathcal{Y}_t}\mathbb{E}_{\rho_t}[\mathbb{E}_{x|y,\rho_t}[\underset{y'\in\mathcal{Y}_t}{\arg\max} \ p_{x|y} \neq \underset{y'\in\mathcal{Y}_t}{\arg\max} \ q_{x|y}]]}_{d_t(p_{x|y},q_{x|y})} \tag{10}$$

Equation 10 describes the disparity $\epsilon_t(\cdot, \cdot)$ from the Bayes-optimal solution by a similarity measure $d_t(\cdot, \cdot)$ in the probability space. Aligning two class-conditionals requires techniques for domain generalization and concept shift. In the future, we will explore efficient class-conditional alignment techniques in the context of online CL.

# B    ALGORITHM

We give the algorithm of Experience Replay in Algorithm 1. The algorithm of our proposed Logit Adjusted Softmax enhanced Experience Replay in Algorithm 2 is mainly based on Algorithm 1. We also apply our method to online GCL by combining with knowledge distillation, as shown in Algorithm 3.

# C    BENCHMARK DETAILS

## C.1    DATASET DETAILS

We list the image size, the total number of training samples, the total number of test samples, and the total number of classes for the 5 datasets (CIFAR10 Krizhevsky (2009), CIFAR100 Krizhevsky (2009), TinyImageNet Le & Yang (2015), ImageNet Deng et al. (2009), and iNaturalist Horn et al. (2017)) in Table 7. In the former four class-balanced datasets, each category contains an

---

**Algorithm 1** Experience Replay (ER) Chaudhry et al. (2019)

---

**Input:** Data stream $\{\mathcal{D}_t\}_{i=1}^T$
**Initialize:** Learner $\Phi(\cdot)$, model parameter $\Theta$, memory buffer $\mathcal{M}_1 \leftarrow \{\}$, label set $\mathcal{Y}_1 \leftarrow \{\}$.
**for** $t = 1$ **to** $T$ **do**
    **Sample incoming batch** $B_t$ **from** $\mathcal{D}_t$
    $\mathcal{Y}_t \leftarrow \mathcal{Y}_{t-1} \cup \text{set}(\{y_i\}_{i=1}^{|B_t|})$
    $B_t^{\mathcal{M}} \leftarrow Retrieval(B_t, \mathcal{M}_t)$
    $z \leftarrow \Phi(\text{concat}(B_t, B_t^{\mathcal{M}}), \Theta)$
    $\text{SGD}(\frac{1}{|B_t|+|B_t^{\mathcal{M}}|} \sum_{i=1}^{|B_t|+|B_t^{\mathcal{M}}|} \mathcal{L}_{\text{CE}}(y_i, z_i), \Theta)$
    $\mathcal{M}_{t+1} \leftarrow Update(B_t, \mathcal{M}_t)$
**end for**

---

---

**Algorithm 2** Experience Replay with Logit Adjusted Softmax (ER-LAS)

---

**Input:** Data stream $\{\mathcal{D}_t\}_{i=1}^T$, temperature scalar $\tau$, sliding window estimator length $l$
**Initialize:** Learner $\Phi(\cdot)$, model parameter $\Theta$, memory buffer $\mathcal{M}_1 \leftarrow \{\}$, label set $\mathcal{Y}_1 \leftarrow \{\}$.
**for** $t = 1$ **to** $T$ **do**
    **Sample incoming batch** $B_t$ **from** $\mathcal{D}_t$
    $\mathcal{Y}_t \leftarrow \mathcal{Y}_{t-1} \cup \text{set}(\{y_i\}_{i=1}^{|B_t|})$
    $B_t^{\mathcal{M}} \leftarrow Retrieval(B_t, \mathcal{M}_t)$
    **for** $y$ **in** $\mathcal{Y}_t$ **do**
        $\pi_{y,t} \leftarrow$ **compute class-priors from Equation 7**
    **end for**
    $z \leftarrow \Phi(\text{concat}(B_t, B_t^{\mathcal{M}}), \Theta)$
    $\text{SGD}(\frac{1}{|B_t|+|B_t^{\mathcal{M}}|} \sum_{i=1}^{|B_t|+|B_t^{\mathcal{M}}|} \mathcal{L}_{\text{LAS}}(y_i, z_i), \Theta)$
    $\mathcal{M}_{t+1} \leftarrow Update(B_t, \mathcal{M}_t)$
**end for**

---

equivalent number of training and test samples. However, within iNaturalist, an inherent imbalance exists between classes, posing a greater challenge. We download the dataset of iNaturalist from `https://github.com/visipedia/inat_comp`.

## C.2   Continual Learning Setup Details

In **online class-IL** (Aljundi et al., 2019), classes of CIFAR10, CIFAR100, TinyImageNet, and ImageNet are evenly split from the total into each task. And the classes in different tasks are disjoint. For example, C-CIFAR10 (5 tasks) is split into 5 disjoint tasks with 2 classes each. As a result, the numbers of training samples, testing samples, and classes are the same in each task, except iNaturalist. We divide the iNaturalist into 26 disjoint tasks according to the initial letter of the category. The numbers of classes in sequential tasks are shown in Figure 3. It shows that the number of classes varies significantly among each task. Noting that the classes within each task are also imbalanced. The comprehensive inter-class imbalance issues of C-iNaturalist (26 tasks) pose great challenges to online CL methods.

In **online GCL** Xie et al. (2022), incoming data contains images from new classes and new domains. We only apply online GCL on CIFAR100. Similar to the online class-IL CL setup, we partition the CIFAR100 dataset into 20 tasks, each with 5 subclasses. However, the model is required to predict superclasses, with each subclass representing a distinct domain within them. Each domain of the superclass has the same number of training samples. As depicted in Figure 4, different superclasses appear in various tasks. Also, varying number of superclasses occur in each task. And the distribution within each superclass changes across different domains. Therefore, G-CIFAR100 (20 tasks) possesses both inter-class imbalance and intra-class domain drift, i.e, changing class-priors and class-conditionals. As one of the most challenging and realistic scenarios for practical applications, it deserves further in-depth investigation in future research.

---

**Algorithm 3** Knowledge distillation with Logit Adjusted Softmax (KD-LAS)

---

**Input:** Data stream $\{\mathcal{D}_t\}_{i=1}^T$, temperature scalar $\tau$, sliding window estimator length $l$
**Initialize:** Learner $\Phi(\cdot)$, model parameter $\Theta$, memory buffer $\mathcal{M}_1 \leftarrow \{\}$, label set $\mathcal{Y}_1 \leftarrow \{\}$.
**for** $t = 1$ **to** $T$ **do**
    Sample incoming batch $B_t$ **from** $\mathcal{D}_t$
    $\mathcal{Y}_t \leftarrow \mathcal{Y}_{t-1} \cup \text{set}(\{y_i\}_{i=1}^{|B_t|})$
    $B_t^{\mathcal{M}} \leftarrow Retrieval(B_t, \mathcal{M}_t)$
    **for** $y$ **in** $\mathcal{Y}_t$ **do**
        $\pi_{y,t} \leftarrow$ **compute class-priors from Equation 7**
    **end for**
    $z \leftarrow \Phi(\text{concat}(B_t, B_t^{\mathcal{M}}), \Theta)$
    $z^{old} \leftarrow \Phi(\text{concat}(B_t, B_t^{\mathcal{M}}), \Theta^{old})$
    $\text{SGD}(\frac{1}{|B_t|+|B_t^{\mathcal{M}}|} \sum_{i=1}^{|B_t|+|B_t^{\mathcal{M}}|} (\mathcal{L}_{\text{LAS}}(y_i, z_i) + \mathcal{L}_{\text{KD}}(z_i, z_i^{old})), \Theta)$
    $\mathcal{M}_{t+1} \leftarrow Update(B_t, \mathcal{M}_t)$
    **if** $t$ **ends a task. then**
        Save the old model $\Theta^{old} \leftarrow \Theta$
    **end if**
**end for**

---

Table 7: Dataset information for CIFAR10, CIFAR100, TinyImageNet, ImageNet, and iNaturalist.

| Dataset | Image Size | # Train | # Test | # Class |
|---|---|---|---|---|
| CIFAR10 Krizhevsky (2009) | $3 \times 32 \times 32$ | 50,000 | 10,000 | 10 |
| CIFAR100 Krizhevsky (2009) | $3 \times 32 \times 32$ | 50,000 | 10,000 | 100 |
| TinyImageNet Le & Yang (2015) | $3 \times 64 \times 64$ | 100,000 | 10,000 | 200 |
| ImageNet Deng et al. (2009) | $3 \times 224 \times 224$ | 1,281,167 | 50,000 | 1000 |
| iNaturalist Horn et al. (2017) | $3 \times 299 \times 299$ | 579,184 | 95,986 | 5089 |

In addition to online class-IL and online GCL, we also consider online blurry continual learning (BCL) to investigate the performance on time-varying long-tail distributions with blurry task boundaries. In **online BCL** Koh et al. (2022), the classes are divided into $N_{\text{blurry}}\%$ disjoint part and $(100 - N_{\text{blurry}}\%)$ blurry part. The classes that belong to the disjoint part will only appear in fixed tasks, while all other classes in the blurry part will occur throughout the data stream. In each task, $(\#\text{train} - (T-1) * M_{\text{blurry}})$ instances will be sampled from the training data of head blurry classes and $M_{\text{blurry}}$ instances will be sampled from the training data of remaining blurry classes, which forms the apparently class-imbalanced blurry part samples. The classes in the blurry part play the role of head classes in turn across different tasks. During inference, the model will predict on test samples from all currently observed classes. We split CIFAR100 and TinyImageNet into 10 blurry tasks according to Koh et al. (2022) with a fixed disjoint ratio $N_{\text{blurry}} = 50$ and blurry level $M_{\text{blurry}} = 10$. Next, we take B-CIFAR100 (10 tasks) as an example. In B-CIFAR100 (10 tasks, $N_{\text{blurry}} = 50$, $M_{\text{blurry}} = 10$, #train = 500 per class, #class = 100), the disjoint part contains 50 classes, and each task possesses 5 disjoint classes of training data. On the other hand, the blurry part comprises the other 50 classes, and each task has 5 head classes. The head classes contain $500 - 9 * 10 = 410$ training samples, whereas the remaining 45 blurry classes only have 10 training samples each for the current task. Therefore, the model in this setup will continuously observe disjoint new classes as stream flows and imbalanced classes overlap across all tasks, encountering a severe problem of inter-class imbalance. We follow Koh et al. (2022) to add the Area Under the Curve of Accuracy $A_{\text{AUC}}$ to evaluate the model performance throughout training in online BCL. We report results on online BCL in Appendix F.8.

## D    METRICS DETAILS

Assume test samples of task $j$ is $S_j = \{x_n, y_n\}_{n=1}^N$. The number of test samples in each class $y$ is $N_y$. The model trained on task $i$ is $\Phi_i$. The seen class set at task $i$ is $\mathcal{Y}_i$. The accuracy $a_{i,j}$ on task $j$

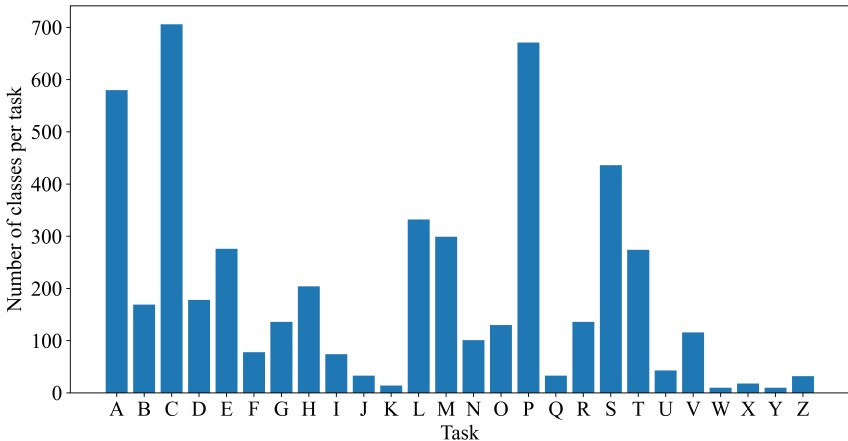

Figure 3: The number of classes per task in divided iNaturalist. Each one of these 26 tasks contains categories with the same corresponding initial letter.

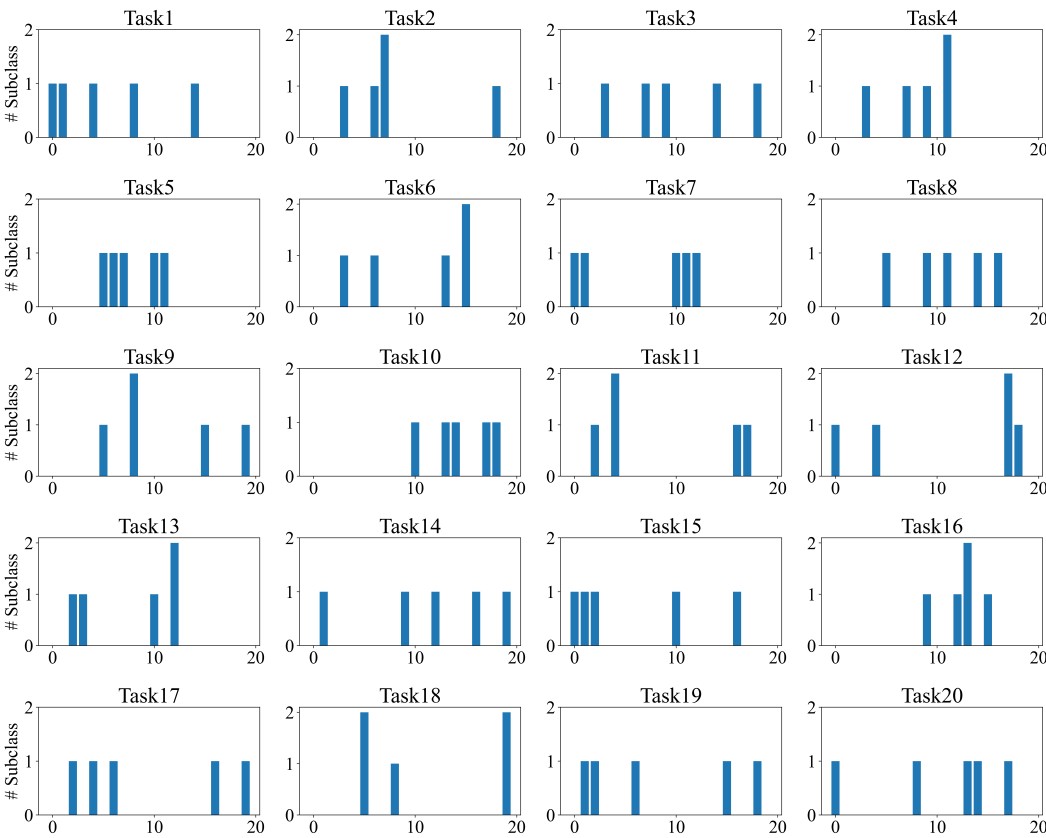

Figure 4: An illustration of the occurrence of subclasses within each superclass for every task in G-CIFAR100 (20 tasks). The $y$-axis represents the number of occurrences of subclasses. The $x$-axis represents the 20 superclasses. Worth noting that each subclass is a distinct domain.

after training on task $i$ is formalized as follows:

$$a_{i,j} = \frac{1}{N} \sum_{n=1}^{N} \mathbb{1}(\arg\max_{y' \in \mathcal{Y}_i} \Phi_{i,y'}(x_n) = y_n). \tag{11}$$

For long-tailed data streams with inherent inter-class imbalance, we also consider a more appropriate metric, namely class-balanced accuracy $a_{i,j}^{\mathrm{cbl}}$, instead of standard accuracy $a_{i,j}$ for evaluation. Class-balanced accuracy excludes prior class imbalances and prevents the overestimation of trivial solutions with high probabilities for major classes.

$$a_{i,j}^{\mathrm{cbl}} = \frac{1}{|\mathcal{Y}_i|} \sum_{y \in \mathcal{Y}_i} \frac{1}{N_y} \sum_{\{(x_n,y_n)|y_n=y\}} \mathbb{1}(\arg\max_{y' \in \mathcal{Y}_i} \Phi_{i,y'}(x_n) = y). \tag{12}$$

The corresponding final average accuracy $A_T$ and final average class-balanced accuracy $A_T^{\mathrm{cbl}}$ can be calculated as follows:

$$A_T = \frac{1}{T} \sum_{j=1}^{T} a_{T,j}, \tag{13}$$

$$A_T^{\mathrm{cbl}} = \frac{1}{T} \sum_{j=1}^{T} a_{T,j}^{\mathrm{cbl}}. \tag{14}$$

The final average forgetting $F_T$ can be computed Chaudhry et al. (2020) as follows:

$$F_T = \frac{1}{T-1} \sum_{j=1}^{T-1} \max_{i \in \{1,...,T-1\}} (a_{i,j} - a_{T,j}). \tag{15}$$

We follow Koh et al. (2022) to add the Area Under the Curve of Accuracy $A_{\mathrm{AUC}}$ in the online blurry CL setup. $A_{\mathrm{AUC}}$ is the average accuracy to {# of samples}. We simplify the calculation of $A_{\mathrm{AUC}}$ by replacing {# of samples} with {# of steps}. Then this metric can be calculated as follows:

$$A_{\mathrm{AUC}} = \frac{1}{N} \sum_{k=1}^{K} f(k \cdot \Delta n) \cdot \Delta n, \tag{16}$$

where $N$ represents the total number of training steps, $\Delta n$ denotes that we sample the accuracy $f(\cdot)$ of the model every $n$ steps, and $K$ is the total number of sample intervals. We set $\Delta n = 5$ in the experiments.

## E  IMPLEMENTATION DETAILS

### E.1  BASELINE IMPLEMENTATION

We as follows list the hyperparameter configurations for the baseline methods mentioned in this paper, along with their sources of code implementation.

For ER (Chaudhry et al., 2019), we set the learning rate as 0.03. The code source is `https://github.com/aimagelab/mammoth`.

For DER++ (Buzzega et al., 2020), we set the learning rate as 0.03. $\alpha$ is set to 0.1, and $\beta$ is set to 0.5. The code source is `https://github.com/aimagelab/mammoth`.

For MRO (Chrysakis & Moens, 2023), we set the learning rate as 0.03. The code source is `https://github.com/aimagelab/mammoth`.

For SS-IL (Ahn et al., 2020), we set the learning rate as 0.03. We update the teacher model every 100 steps. The code source is `https://github.com/hongjoon0805/SS-IL-Official`.

For CLIB (Koh et al., 2022), we set the learning rate as 0.03. The period between sample-wise importance updates is set to 3. The code source is `https://github.com/naver-ai/i-Blurry`.

For ER-ACE (Caccia et al., 2022), we set the learning rate as 0.03. The code source is `https://github.com/pclucas14/AML`.

For ER-OBC (Chrysakis & Moens, 2023), we set the learning rate as 0.03 for both training and bias correction. The code source is `https://github.com/chrysakis/OBC`.

For MIR (Aljundi et al., 2019), we set the learning rate as 0.03. The number of subsampling in replay is 160. The code source is `https://github.com/optimass/Maximally_Interfered_Retrieval`.

For ASER$_\mu$ (Shim et al., 2020), we set the learning rate as 0.03. The number of nearest neighbors $K$ to perform ASER is 5. We use mean values of adversarial Shapley values and cooperative Shapley values. The maximum number of samples per class for random sampling is 6.0 times of incoming batch size. The code source is `https://github.com/RaptorMai/online-continual-learning`.

For OCS (Yoon et al., 2022), we set the learning rate as 0.03. The hyperparameter $\tau$ that controls the degree of model plasticity and stability is set to 1000.0. The code source is `https://openreview.net/forum?id=f9D-5WNG4Nv`.

For LwF (Li & Hoiem, 2016), we set the learning rate as 0.03. The penalty weight $\alpha$ is set to 0.5 and the temperature scalar is set to 2.0. The code source is `https://github.com/aimagelab/mammoth`.

For LURIC (Hou et al., 2019), we set the learning rate as 0.03. $\lambda_{\text{base}}$ is set to 5.0 for all the experiments. The code source is `https://github.com/hshustc/CVPR19_Incremental_Learning`.

For GeoDL (Simon et al., 2021), we set the learning rate as 0.03. The adaptive weight $\beta$ is set to 5.0. The code source is `https://github.com/chrysts/geodesic_continual_learning`.

We also list the hyperparameter configuration for the baseline methods used in this appendix with their sources of code implementation.

For SCR (Mai et al., 2021), we set the learning rate as 0.03 and the temperature as $\tau = 0.07$. The code source is `https://github.com/RaptorMai/online-continual-learning`.

For OCM (Guo et al., 2022), we use Adam optimizer and set the learning rate as 0.001. The code source is `https://github.com/gydpku/OCM`.

For BiC (Wu et al., 2019), we set the learning rate as 0.03. We split 10% of the training data into a validation set for training the bias injector with 50 epochs. The softmax temperature $T$ is 2.0. Distillation loss is also applied after bias correction. The code source is `https://github.com/sairin1202/BIC`.

For E2E(Castro et al., 2018), we set the learning rate as 0.03. In the process of balanced fine-tuning, we set the learning rate as 0.003 and train 30 epochs. The code source is `https://github.com/PatrickZH/End-to-End-Incremental-Learning`.

For IL2M(Belouadah & Popescu, 2019), we set the learning rate as 0.03. We calculate the mean and variance of each batch online to re-scale the outputs. The code source is `https://github.com/EdenBelouadah/class-incremental-learning`.

For LUCIR (Hou et al., 2019), we set the learning rate as 0.03. $\lambda_{\text{base}}$ is set to 5.0, $K$ is set to 2, and $m$ is set to 0.5 for all the experiments. The code source is `https://github.com/hshustc/CVPR19_Incremental_Learning`.

### E.2 ABLATION IMPLEMENTATION

In the ablation study of §6.3, we employ two extreme variants of LAS, along with two estimators. *Random* estimator randomly assigns class-priors. *Macro* uses statistical information to assign class-priors. Now, we elaborate on how these four methods are implemented. Recalling our proposed Logit Adjusted Softmax cross-entropy loss in Equation 6.

$$\mathcal{L}_{\text{LAS}}(y, \Phi(x)) = -\log \frac{e^{\Phi_y(x)+\tau \cdot \log \pi_{y,t}}}{\sum_{y' \in \mathcal{Y}_t} e^{\Phi_{y'}(x)+\tau \cdot \log \pi_{y',t}}} = \log[1 + \sum_{y' \neq y} \left(\frac{\pi_{y',t}}{\pi_{y,t}}\right)^\tau \cdot e^{\left(\Phi_{y'}(x)-\Phi_y(x)\right)}].$$

(17)

$\tau = 0$ is a simple special case that sets the temperature scalar to 0.

$\tau = \infty$ needs modification because directly setting the hyperparameter $\tau$ to a large value to pursue $\infty$ would cause troubles when $\pi_{y',t}/\pi_{y,t} > 1$, as it would lead to an infinity coefficient and result in gradient explosion, obstructing the gradient descent optimization algorithm. Therefore, as shown in Equation 18, we set the coefficient to 0 only when $\pi_{y',t}/\pi_{y,t} < 1$, while retaining $\tau = 1$ for all other situations to enable successful model training. The significantly low forgetting rate and competitive accuracy observed in the experimental results suggest that this approach closely approximates $\tau = \infty$ as expected.

$$(\pi_{y',t}/\pi_{y,t})^\tau = \begin{cases} 0, & (\pi_{y',t}/\pi_{y,t}) < 1 \\ (\pi_{y',t}/\pi_{y,t}), & \text{otherwise} \end{cases}. \tag{18}$$

*Random* samples each prior of seen classes from a uniform distribution $U[0,1]$. Then they are normalized to 1.

*Macro* computes the joint label distribution by taking into account the occurrence frequencies of each class in the current data stream, as well as the label probabilities in the memory buffer, to serve as the current class-priors. It is worth noting that since the distribution of the data stream is unknown during training, *Macro* cannot be directly obtained and serves only as a reference for comparing and validating the necessity of batch-wise estimators. For instance, in C-CIFAR (5 tasks), when it comes to the 2$^{\text{nd}}$ task, 2 classes in the data stream are of the same quantity, and the 2 classes in the memory buffer also contain a similar number of samples from the previous task. The incoming and buffer batch sizes are also the same. At this point, the 4 classes probabilities that may appear in the input batch are all equally likely, i.e., $1/4$. When it comes to the 5$^{\text{th}}$ task, the data stream still consists of 2 classes with the same label probabilities, but the memory buffer now stores 8 classes that have appeared before. Therefore, it can be calculated that the class-priors of the 2 classes in the data stream are $1/4$, while the class-priors of the 8 classes in the memory buffer are $1/16$. *Macro* represents a statistical oracle, but experiments show that its performance is inferior to batch-wise estimators, indicating that in online CL, the model may pay more attention to the label distributions within each batch rather than the label distributions across the sequential tasks.

## F  MORE EXPERIMENTAL RESULTS

### F.1  RESULTS WHEN BATCH SIZES ARE VARIED

While typically samples arrive one by one in the online learning data stream, advanced algorithms(Caccia et al., 2022; Chrysakis & Moens, 2023) are commonly designed to update the model by accumulating a certain number of incoming samples as a batch. This is because per-batch updating is generally more advantageous for model optimization convergence and well-defined classification boundaries than updating on each individual sample. However, in some situations with constrained computational resources, only very small batch sizes are available or the batch sizes vary. Based on this concern, we consider two setups related to changing the batch size: one is various batch sizes for the entire online training process, and the other is varying the batch size during training. We conduct experiments on online C-CIFAR10. We begin with brief introductions to these two setups.

1. Evaluating the batch size change for the entire online training process examines the macro robustness of our method to the hyperparameter of batch size. In the manuscript, we set both incoming and buffer batch sizes to 32. We now experiment with corresponding batch sizes of 4, 16, 64. Smaller batch sizes bring more gradient updates for the model, but each contains less information for forming inter-class margins. Larger batch sizes may lead to overfitting on the memory buffer, thereby reducing performance.

2. Varying the batch size throughout the entire online training process examines the micro robustness of the batch size. This is a practical scenario where the frequency of incoming data may vary at different stages, requiring time-varying batch sizes. In this experiment, we only vary incoming batch sizes while keeping buffer batch sizes at 32. We consider 3 cases of changing incoming batch sizes:

   - Increasing incoming batch sizes during training, specifically for C-CIFAR10: 2, 4, 8, 16, 32, as **Increase**. The inter-class imbalance issue intensifies.

- Decreasing incoming batch sizes during training, specifically for C-CIFAR10: 32, 16, 8, 4, 2, as **Decrease**. The inter-class imbalance issue is alleviated.
- Randomly sampling incoming batch sizes from a uniform distribution $U[2, 32]$ at each stage, as **Random**. This is a fusion of the previous two cases, where the impact of inter-class imbalance varies during training.

Table 8: Comparison of final average accuracy on online C-CIFAR10 with various batch sizes. In the manuscript, we set both incoming and buffer batch sizes to 32. We experiment with corresponding batch sizes of 4, 16, and 64. Experimental settings are the same as in Table 1. Memory sizes are $M = 1k$.

| Batch size | 4 | 16 | 32 | 64 |
|---|---|---|---|---|
| ER | $54.0 \pm 1.8$ | $52.4 \pm 1.8$ | $45.4 \pm 1.8$ | $45.4 \pm 1.9$ |
| ER-ACE | $55.1 \pm 1.9$ | $56.7 \pm 2.1$ | $48.1 \pm 1.1$ | $46.2 \pm 1.9$ |
| ER-OBC | $48.6 \pm 1.8$ | $54.7 \pm 1.4$ | $46.4 \pm 0.6$ | $39.0 \pm 1.8$ |
| ER-LAS | $\mathbf{59.2 \pm 1.2}$ | $\mathbf{57.5 \pm 1.3}$ | $\mathbf{55.3 \pm 1.6}$ | $\mathbf{53.1 \pm 1.2}$ |

Table 9: Comparison of final average accuracy on online C-CIFAR10 with varying batch sizes during training. We only vary incoming batch sizes while keeping buffer batch sizes as 32: **Increase** incoming batch sizes during training, i.e., 2, 4, 8, 16, 32. **Decrease** incoming batch sizes during training, i.e., 32, 16, 8, 4, 2. **Random**ly sampling incoming batch sizes from a uniform distribution $U[2, 32]$ at each stage. Experimental settings are the same as in Table 1. Memory sizes are $M = 1k$.

| Incoming batch size | Increase | Decrease | Random |
|---|---|---|---|
| ER | $52.7 \pm 1.8$ | $65.2 \pm 1.9$ | $55.1 \pm 1.8$ |
| ER-ACE | $50.8 \pm 1.2$ | $62.5 \pm 1.1$ | $55.1 \pm 1.9$ |
| ER-OBC | $53.1 \pm 1.4$ | $65.3 \pm 1.1$ | $51.4 \pm 1.7$ |
| ER-LAS | $\mathbf{59.8 \pm 1.1}$ | $\mathbf{65.7 \pm 0.7}$ | $\mathbf{59.3 \pm 1.9}$ |

The results in Table 8 and Table 9 show that ER-LAS consistently achieves the best accuracy across various and varying batch sizes, highlighting the robustness of our method to batch size variations. In theory, changing batch sizes or the variation of batch sizes during training poses no threat to our principle of mitigating inter-class imbalance through the elimination of class-priors. It only affects our estimation of time-varying class-priors. However, the ablation study of estimators in §6.3 indicates that online CL models may pay more attention to the current input class distributions. Therefore, our designed batch-wise estimator can timely provide effective approximation at various batch sizes. The potential issues may lie in the cases where batch sizes become extremely small, such as 1. Following, we discuss this problem in detail and provide recommendations for improvement.

In fact, training on a single incoming sample goes against the theory of traditional stochastic gradient descent, which may harm model convergence and hinder the establishment of classification boundaries. Therefore, we maintain the incoming batch size of 1 and consider concatenating various numbers of buffer batch sizes to ensure valid training and practical performance. The experiments are conducted on online C-CIFAR10 in order to explore the impact of changed buffer batch sizes on a single incoming batch size.

Table 10: Comparison of final average accuracy on online C-CIFAR10 with fixed incoming batch sizes of 1 and various buffer batch sizes. Experimental settings are the same as in Table 1. Memory sizes are $M = 1k$.

| Buffer batch size | 1 | 4 | 16 | 64 |
|---|---|---|---|---|
| ER | $\mathbf{39.3 \pm 2.0}$ | $62.4 \pm 2.0$ | $63.7 \pm 1.8$ | $60.6 \pm 1.9$ |
| ER-ACE | $27.9 \pm 2.1$ | $57.7 \pm 1.7$ | $58.1 \pm 1.8$ | $54.5 \pm 1.7$ |
| ER-OBC | $33.2 \pm 1.9$ | $64.3 \pm 1.8$ | $65.3 \pm 1.8$ | $60.9 \pm 1.7$ |
| ER-LAS | $36.9 \pm 1.5$ | $\mathbf{66.4 \pm 1.5}$ | $\mathbf{67.2 \pm 1.5}$ | $\mathbf{62.2 \pm 1.3}$ |

The results in Table 10 show that when both incoming and buffer batch sizes are 1, ER-LAS performs slightly worse than the ER baseline. Nevertheless, simply increasing buffer batch sizes can enable ER-LAS to achieve the highest accuracy. This is because the case of extremely small batch sizes of 1 affects our estimation of time-varying class-priors and hinders the construction of classification margins, where slightly increasing buffer batch sizes can serve as an effective approach to refresh our method. Noting that excessive buffer batch sizes can lead to overfitting on the memory buffer and harm performance, as shown in the rightmost column of Table 10.

## F.2 COMPARISON TO INTER-CLASS IMBALANCE MITIGATION METHODS

We mentioned the differences between LAS and other class imbalance mitigation methods from an analytical perspective in §5. We worry that since these methods have not been deliberately designed and applied to online CL in previous works, our direct application may lack credibility and endorsement. As a result, we do not compare with these methods in experiments. However, we have indeed conducted experiments with them in the preliminary exploration phase of our method. Here, we briefly describe our applications and provide experimental comparisons and analysis. We compare four class imbalance mitigation methods originally for stable distributions. We refer to Cui et al. (2019) and apply the Class-Balanced loss, which re-weights the loss terms of each class based on the input class distribution, as **ER-CBL** of Loss weighting. We refer to Kang et al. (2020) and normalize the weights of classifiers with $\|w_y\|_2$, as **ER-WN** of Weight normalization. We perform upsampling on the buffer samples and downsampling by randomly ignoring some incoming samples to maintain consistent input class distributions, as **ER-Up** and **ER-Down** of Resampling (Kubát & Matwin, 1997).

Table 11: Comparison of final average accuracy by ER, ER-LAS, and imbalance mitigation methods. Experimental settings are the same as in Table 1. Memory sizes are $M = 1k$.

| Dataset | C-CIFAR10 | C-CIFAR100 | C-TinyImageNet |
|---|---|---|---|
| ER | $45.4 \pm 1.8$ | $16.5 \pm 0.4$ | $11.0 \pm 0.2$ |
| ER-CBL | $48.1 \pm 1.6$ | $18.8 \pm 0.2$ | $11.1 \pm 0.2$ |
| ER-WN | $46.1 \pm 1.5$ | $16.6 \pm 0.8$ | $11.0 \pm 0.1$ |
| ER-Up | $53.6 \pm 1.6$ | $23.4 \pm 0.5$ | $15.0 \pm 0.1$ |
| ER-Down | $48.2 \pm 1.6$ | $18.9 \pm 0.3$ | $13.4 \pm 0.2$ |
| ER-LAS | $\mathbf{55.3 \pm 1.6}$ | $\mathbf{25.7 \pm 0.3}$ | $\mathbf{15.5 \pm 0.2}$ |

The results in Table 11 show that our ER-LAS outperforms all other compared methods for mitigating class imbalance. Next, we will analyze the shortcomings of these methods. ER-CBL re-weights the loss after computing the logits and ground truth, which helps in learning better features of minority classes but fails to eliminate the influence of class-priors to achieve balanced posterior outputs. ER-WN ensures that the model output is not affected by class weight bias. However, we find that CL models are still affected by feature drift(Caccia et al., 2022), leading to recency bias. Therefore, these two methods cannot truly solve the inter-class imbalance problem in online CL. ER-Up is the closest method to our ER-LAS, but as the inter-class imbalance problem intensifies, it results in a significant computational burden, whereas our method costs almost no additional computational resources. ER-Down, although maintaining inter-class balance during training, discards a majority of valuable incoming training samples. Furthermore, unlike these four methods, our proposed LAS is supported by a statistical ground of underlying class-conditionals.

## F.3 RESULTS ON OFFLINE TASK-IL SCENARIOS

In offline task-IL settings, learners have access to a whole dataset for each task and can undergo multiple epochs of training. Previous arts that are highly related to our work have proposed some Logit Rectify methods to alleviate the issue of inter-class imbalance in offline CL and improve learning performance. BiC (Wu et al., 2019) adds a bias correction layer to the model and stores a portion of input data as the held-out validation set to calibrate this layer and lessen the model's task-recency bias. E2E (Castro et al., 2018) fine-tunes the model with a balanced dataset after each task. IL2M (Belouadah & Popescu, 2019) rescales the model output with historical statistics. LURIC (Hou et al., 2019) combines cosine normalization, less-forget constraint, and inter-class

separation to mitigate the adverse effects of class imbalance. We also compare successful offline CL methods (ER (Chaudhry et al., 2019), DER++ (Buzzega et al., 2020), and ER-ACE (Caccia et al., 2022)). Our experimental settings follow (Buzzega et al., 2020). LUCIR fails to work on C-TinyImageNet due to too low memory size.

Table 12: Final average accuracy $A_T$ (higher is better) on C-CIFAR10 (5 tasks), C-CIFAR100 (10 tasks), and C-TinyImageNet (10 tasks) in the offline condition. $M = 100$. The epoch is set to 50.

| Dataset | C-CIFAR10 | C-CIFAR100 | C-TinyImageNet |
|---|---|---|---|
| BiC | $23.4_{\pm 0.8}$ | $15.3_{\pm 0.1}$ | $10.1_{\pm 0.1}$ |
| E2E | $51.6_{\pm 0.3}$ | $16.7_{\pm 0.1}$ | $9.0_{\pm 0.0}$ |
| IL2M | $42.1_{\pm 0.6}$ | $11.0_{\pm 0.2}$ | $8.4_{\pm 0.1}$ |
| LUCIR | $28.9_{\pm 1.0}$ | $15.7_{\pm 0.7}$ | $10.2_{\pm 0.1}$ |
| ER | $39.4_{\pm 0.3}$ | $11.5_{\pm 0.1}$ | $8.1_{\pm 0.0}$ |
| DER++ | $55.3_{\pm 1.2}$ | $14.8_{\pm 1.8}$ | $9.4_{\pm 0.3}$ |
| ER-ACE | $55.9_{\pm 1.0}$ | $17.7_{\pm 0.7}$ | $8.7_{\pm 0.2}$ |
| ER-LAS | $53.9_{\pm 1.0}$ | $16.4_{\pm 0.2}$ | $10.3_{\pm 0.1}$ |
| LwF-LAS | $\mathbf{57.5_{\pm 0.2}}$ | $\mathbf{22.6_{\pm 0.1}}$ | $\mathbf{12.4_{\pm 0.1}}$ |

The results in Table 12 demonstrate that ER-LAS still achieves competitive performance in offline CL. When combined with knowledge distillation method LwF (Li & Hoiem, 2016), LwF-LAS outperform the other compared methods. These finds indicate that our approach can also effectively mitigate inter-class imbalance and improve performance than previously proposed Logit Rectify methods in offline task-IL setups. Considering the severe impact of recency bias in online CL, our main focus is how to eliminate the adverse effects caused by inter-class imbalance in online settings, and we design a widely applicable LAS algorithm.

## F.4 COMPARISON TO CL METHODS BASED ON CONTRASTIVE LEARNING

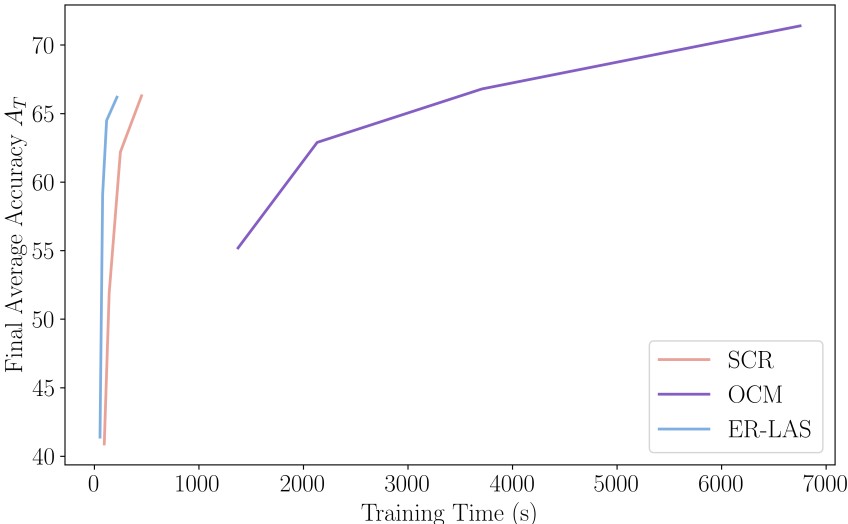

Figure 5: Comparison with online CL methods based on contrastive learning on C-CIFAR10 (5 tasks). Memory size $M = 1k$. The $x$-axis represents training time, and the $y$-axis represents the final average accuracy $A_T$ (higher is better). We evaluate the accuracy and the time efficiency of SCR, OCM, and our ER-LAS at batch sizes of 8, 16, 32, and 64. Noting that the time consumption increases as the batch size decreases.

We compare our method with the online CL methods (SCR Mai et al. (2021) and OCM Guo et al. (2022)) based on contrastive learning. SCR utilizes the NCM classifier and is trained via supervised contrastive learning. OCM employs contrastive learning to maximize mutual information. These methods based on contrastive learning typically require more computational resources, and

their performance is influenced by the number of negative samples, but they often achieve better performance. As shown in Figure 5, we evaluate the training time and the final average accuracy of our ER-LAS and the contrastive learning-based online CL methods under different batch sizes. SCR and OCM require 2x and 30x more computational time than our method, respectively. Although they achieve higher accuracy than our method, LAS exhibits superior overall computational efficiency.

## F.5 FORGETTING RATE

Table 13: Final average forgetting $F_T$ (lower is better) on C-CIFAR10 (5 tasks), C-CIFAR100 (10 tasks), and C-TinyImageNet (10 tasks). $M$ is the memory buffer size.

| Dataset | C-CIFAR10 | | | C-CIFAR100 | | | C-TinyImageNet | | |
|---|---|---|---|---|---|---|---|---|---|
| Method | $M = 0.5k$ | $M = 1k$ | $M = 2k$ | $M = 0.5k$ | $M = 1k$ | $M = 2k$ | $M = 0.5k$ | $M = 1k$ | $M = 2k$ |
| ER | $43.0_{\pm1.5}$ | $36.2_{\pm1.6}$ | $24.5_{\pm1.3}$ | $31.1_{\pm0.6}$ | $23.2_{\pm1.0}$ | $23.2_{\pm0.6}$ | $38.5_{\pm0.5}$ | $33.4_{\pm0.3}$ | $27.8_{\pm0.3}$ |
| DER++ | $29.3_{\pm1.2}$ | $31.6_{\pm2.9}$ | $32.4_{\pm2.3}$ | $37.6_{\pm0.5}$ | $34.5_{\pm0.5}$ | $36.4_{\pm0.6}$ | $38.6_{\pm0.2}$ | $37.2_{\pm0.3}$ | $37.2_{\pm0.2}$ |
| MRO | $26.1_{\pm1.5}$ | $21.0_{\pm1.2}$ | $8.9_{\pm0.8}$ | $13.5_{\pm0.3}$ | $9.3_{\pm0.3}$ | $6.3_{\pm0.2}$ | $\mathbf{11.1_{\pm0.1}}$ | $10.9_{\pm0.2}$ | $8.4_{\pm0.2}$ |
| SS-IL | $22.0_{\pm0.8}$ | $20.0_{\pm0.9}$ | $16.5_{\pm0.5}$ | $11.8_{\pm0.3}$ | $10.0_{\pm0.2}$ | $8.1_{\pm0.3}$ | $14.5_{\pm0.2}$ | $12.2_{\pm0.2}$ | $10.0_{\pm0.8}$ |
| CLIB | $30.4_{\pm1.6}$ | $17.7_{\pm1.5}$ | $16.1_{\pm1.3}$ | $25.9_{\pm0.3}$ | $14.9_{\pm0.3}$ | $7.6_{\pm0.3}$ | $30.1_{\pm0.3}$ | $20.4_{\pm0.3}$ | $10.8_{\pm0.3}$ |
| ER-ACE | $\mathbf{11.0_{\pm1.6}}$ | $\mathbf{16.1_{\pm1.3}}$ | $10.1_{\pm1.3}$ | $\mathbf{9.3_{\pm0.7}}$ | $\mathbf{7.9_{\pm0.5}}$ | $\mathbf{5.6_{\pm0.7}}$ | $13.8_{\pm0.3}$ | $\mathbf{9.9_{\pm0.3}}$ | $\mathbf{7.5_{\pm0.4}}$ |
| ER-OBC | $37.3_{\pm0.8}$ | $19.5_{\pm0.6}$ | $30.5_{\pm0.8}$ | $24.0_{\pm0.3}$ | $22.4_{\pm0.4}$ | $18.7_{\pm0.4}$ | $36.2_{\pm0.2}$ | $29.4_{\pm0.2}$ | $21.9_{\pm0.2}$ |
| ER-LAS | $28.5_{\pm1.3}$ | $18.5_{\pm1.4}$ | $\mathbf{7.1_{\pm1.1}}$ | $22.1_{\pm0.4}$ | $11.5_{\pm0.6}$ | $9.3_{\pm0.7}$ | $26.9_{\pm0.2}$ | $19.9_{\pm0.4}$ | $10.3_{\pm0.2}$ |

We compare the forgetting rate of each method on C-CIFAR10, C-CIFAR100, and C-TinyImageNet in Table 13. In most settings, ER-ACE achieved the lowest forgetting rate, except for when compared to our proposed ER-LAS on C-CIFAR10 with $M = 2k$, and to MRO on C-TinyImageNet with $M = 0.5k$. Noting that the lowest forgetting rate does not necessarily correspond to the highest accuracy. Moreover, remarkable reductions in the forgetting rate can be achieved by deliberately adjusting the hyperparameters of our method, but at the cost of accuracy. Currently, our method achieves the optimal stability-plasticity trade-off.

## F.6 PREDICTION RESULTS ON C-IMAGENET

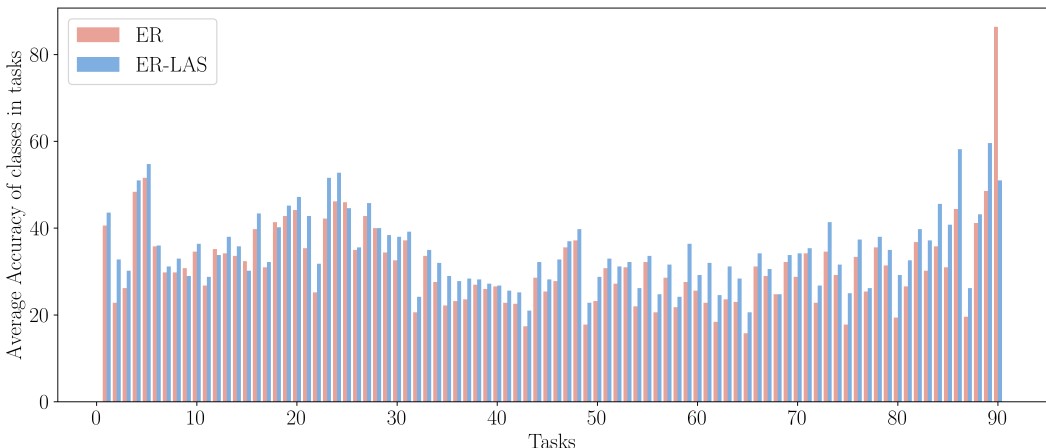

Figure 6: Prediction results by ER and ER-LAS on C-ImageNet (90 tasks). We calculate the average accuracy of classes within each task to demonstrate the recency bias.

We present the prediction results of ER and our proposed ER-LAS on C-ImageNet after training, as shown in Figure 6. Recalling that ER assigns 38% of the test samples to the most recently learned classes in C-CIFAR100 (Figure 1 in §4). ER also outperforms ER-LAS on the last task, but is inferior to ER-LAS on all other tasks. This is due to the larger task sequence and the more total number of classes in C-ImageNet than in C-CIFAR100, resulting in a much more severe recency bias for the ER method. However, our ER-LAS successfully eliminates the recency bias as expected, and as a result, achieves a remarkably lower forgetting rate and the highest accuracy in the experiments of

§6.1. These results validate that inter-class imbalance is more severe in long sequential tasks and demonstrate that our method can adapt to learning from such highly imbalanced data streams by pursuing the class-conditional function.

### F.7 Class-balanced Accuracy on C-iNaturalist

Table 14: Final average accuracy $A_T$ and final average class-balanced accuracy $A_T^{cbl}$ (both higher is better) on C-iNaturalist (26 tasks). We show the results of top-3 methods. Memory sizes are $M = 20k$.

| Dataset | iNaturalist | |
| --- | --- | --- |
| **Method** | $A_T \uparrow$ | $A_T^{cbl} \uparrow$ |
| ER | $4.66_{\pm 0.01}$ | $6.25_{\pm 0.01}$ |
| ER-ACE | $5.68_{\pm 0.01}$ | $6.32_{\pm 0.01}$ |
| MRO | $4.96_{\pm 0.0}$ | $4.47_{\pm 0.01}$ |
| ER-LAS | $\mathbf{8.11}_{\pm \mathbf{0.01}}$ | $\mathbf{8.62}_{\pm \mathbf{0.01}}$ |

As we aim to pursue the Bayes-optimal classifier that minimizes the class-balanced error on imbalanced data streams, we also evaluate the class-balanced accuracy of our method and baselines on C-iNaturalist. As shown in Table 14, ER-LAS[†] achieves the best performance in both accuracy and class-balanced accuracy, validating the effectiveness of our optimization towards the Bayes-optimal estimator.

### F.8 Results on Blurry CL Scenarios

Online blurry CL (B) (Koh et al., 2022) has both class-IL distributions and blurry task boundaries. It divides the classes into $N_{blurry}\%$ disjoint part and $(100 - N_{blurry}\%)$ blurry part. The disjoint part classes only appear in fixed tasks, while the blurry part classes occur throughout the data stream but with inherent inter-class imbalance represented by blurry level $M_{blurry}$. We split CIFAR100 and TinyImageNet into 10 blurry tasks according to Koh et al. (2022) with a disjoint ratio $N_{blurry} = 50$ and blurry level $M_{blurry} = 10$. We compare ER-LAS with the best 3 baselines on B-CIFAR100 and B-TinyImageNet.

Table 15: AUC of Accuracy $A_{AUC}$ and final average accuracy $A_T$ (both higher is better) on B-CIFAR100 (10 tasks) and B-TinyImageNet (10 tasks) with disjoint ratio $N_{blurry} = 50$ and blurry level $M_{blurry} = 10$. We show the results of top-3 methods. Memory sizes are $M = 2k$.

| Dataset | B-CIFAR100 | B-TinyImageNet |
| --- | --- | --- |
| **Method** | $A_T \uparrow / A_{AUC} \uparrow$ | $A_T \uparrow / A_{AUC} \uparrow$ |
| ER | $19.6_{\pm 1.6}/16.1_{\pm 0.1}$ | $16.2_{\pm 0.2}/12.4_{\pm 0.0}$ |
| ER-ACE | $18.3_{\pm 1.0}/15.2_{\pm 0.0}$ | $16.4_{\pm 0.3}/12.2_{\pm 0.1}$ |
| CLIB | $21.9_{\pm 0.3}/18.0_{\pm 0.1}$ | $15.9_{\pm 0.2}/12.6_{\pm 0.1}$ |
| ER-LAS | $\mathbf{24.9}_{\pm \mathbf{0.5}}/\mathbf{20.3}_{\pm \mathbf{0.0}}$ | $\mathbf{19.4}_{\pm \mathbf{0.4}}/\mathbf{15.1}_{\pm \mathbf{0.0}}$ |

Table 15 shows that ER-LAS can outperform all baselines on $A_T$ and $A_{AUC}$. For example, ER-LAS improves the best baseline by 3.0% $A_T$ and 2.5% $A_{AUC}$ on B-TinyImageNet. In fact, our method is particularly suitable for the online blurry CL setup because LAS alleviates the detrimental effects of inter-class imbalance both inherently in the data stream and between new and old classes. The results of ER-LAS further confirm that such an advantage can help obtain high accuracy throughout learning under the realistic online blurry CL setup with challenging inter-class imbalance problems.

