# OpenReview forum: "Online Continual Learning via Pursuing Class-conditional Funtion"
_ICLR.cc/2024/Conference — ICLR 2024 Conference Withdrawn Submission_

### Official Review · Reviewer_MqmM · 2023-10-29

**Soundness:** 3 good
**Presentation:** 2 fair
**Contribution:** 3 good
**Rating:** 5
**Confidence:** 4

**Summary:**

- The authors theoretically analyze that inter-class imbalance is entirely attributed to imbalanced class priors and the class-conditional function learned from intra-class distributions is the Bayes-optimal classifier.
- Accordingly, The authors present that a simple adjustment of model logits during training can effectively resist prior class bias and grasp the corresponding Bayes-optimum.
- The proposed method mitigates the impact of inter-class imbalance not only in class-incremental but also in realistic general setups by eliminating class priors and pursuing class conditionals.

**Strengths:**

- (+) The proposed method mitigates the impact of inter-class imbalance not only in class increments but also in realistic general settings by removing class priors and pursuing class conditions.
- (+) The extensive experiments demonstrate the effectiveness of the proposed method on various benchmarks and show significant performance improvements.

**Weaknesses:**

- (-) There is no interpretation between the imbalanced ratio and performance in online settings.
- (-) Prior class-imbalanced baseline references seem insufficient, such as LDAM[1], Maximum Margin[2], and Balanced Softmax[3]. Minorly, the proposed method looks similar to the Maximum Margin [2] class imbalanced dataset learning in Sec. 4.2 logit adjusted softmax cross-entropy loss.
    - [1] Learning Imbalanced Datasets with Label-Distribution-Aware Margin Loss
    - [2] Learning Imbalanced Datasets With Maximum Margin Loss
    - [3] Balanced Meta-Softmax for Long-Tailed Visual Recognition
- (-) In the real world, sometimes, no task boundary is given, but task Incremental setting is also crucial in continual learning. Comparison of prior architecture-based methods seems insufficient, such as SUPSUP[4] and WSN[5] in task incremental setting.
    - [4] Forget-free Continual Learning with Winning Subnetworks
    - [5] Supermasks in superposition

**Questions:**

- What advantages does the proposed method have over existing methods regarding imbalanced learning above references? Some depend on imbalanced class priors, but they could also be good baselines to strengthen the proposed method.
- How does the proposed learning method contribute in task incremental setting (task-boundaries given) when applied to the architecture-based methods?

**Details Of Ethics Concerns:**

None.

---

### Official Review · Reviewer_Y84H · 2023-10-31

**Soundness:** 2 fair
**Presentation:** 3 good
**Contribution:** 2 fair
**Rating:** 3
**Confidence:** 4

**Summary:**

This paper studies the online continual learning problem, which learns from non-stationary data stream in a single-pass manner. The authors identify the class imbalance issue in continual learning, and propose to use Logit Adjustment to address the imbalance issue. The effectiveness of the proposed method is verified by benchmark experiments.

**Strengths:**

1. This paper is generally well-written, and the organization is clear.
2. Leveraging logit adjustment in online class-incremental learning and class-priors estimation are well-motivated, simple to implement and easy to follow.
3. Extensive experiments demonstrated the performance gain of addressing the class imbalance problem in online continual learning.

**Weaknesses:**

1. Despite the effectiveness, the logit adjustment (Menon et al. 2021) technique is well-known in long-tailed recognition, thus raises the concern of lacking novelty.

2. Lack of understanding of how the proposed method (LAS) influences the stability and plasticity of continual learners. For example, LAS may remarkably reduce the forgetting of old classes (improve stability) but also, with reduced plasticity, may sacrifice the accuracy of new classes significantly.

3. There are some issues that need to be clarified. In section 6.2, experiments are also conducted on “the most difficult and realistic online GCL setup”. However, as shown in Table 4 and Table 3, the performance (e.g., ER and ER+LAS) under online GCL setup is much better than that of online CIL, which seems to imply that the online GCL is simpler. Besides, it is known that LUCIR and GeoDL are much stronger than LwF in offline CIL. However, in Table 4, LwF performs remarkably better than LUCIR and GeoDL. The reviewer suggests authors include further clarification about those points.

**Questions:**

See weaknesses.

---

### Official Review · Reviewer_3g12 · 2023-11-01

**Soundness:** 3 good
**Presentation:** 3 good
**Contribution:** 2 fair
**Rating:** 5
**Confidence:** 4

**Summary:**

This paper addresses the challenge of online continual learning by tackling inter-class imbalance problem. The author uses a method that adjusts model logits during training to resist prior class bias, aiming for the Bayes-optimal classifier based on class-conditional functions, with minimal additional computational cost. Experimental results show significant performance improvements compared to previous methods.

**Strengths:**

1. The presentation of the paper is clear, and the experiments are highly detailed.

2. The author provides some theoretical results to elucidate their motivation.

**Weaknesses:**

1. This paper's method seems to be a simple combination of Logit Adjustment [Menon et al. 2021] and online continual learning, with a similar organization and no additional insights. Although the authors propose using sliding window estimates for priors in an online environment, in my opinion, the contribution of this paper is somewhat limited.

**Questions:**

Please refer to [Weaknesses].

---

### Official Review · Reviewer_6ngC · 2023-11-01

**Soundness:** 2 fair
**Presentation:** 2 fair
**Contribution:** 2 fair
**Rating:** 3
**Confidence:** 4

**Summary:**

This paper first pointed out that the inter-class imbalance is highly attributed to imbalanced class prior, and then proposed the time-varying class prior to adaptively reflect the statistics of seen class labels. Furthermore, by utilizing the Logit Adjustment Softmax (LAS) cross-entropy loss, the proposed model can prevent the prediction bias. The authors show theoretical results that using the class-conditional function can minimize the class-balanced error. In the experiment, the proposed method outperforms the state-of-the-art baselines

**Strengths:**

1. Different from previous methods (e.g. using knowledge distillation or separated softmax), the proposed method is more adaptive to the situation in which the class statistics is constantly changing.

**Weaknesses:**

1. In my opinion, the contribution is somewhat minor. First, the motivation behind using time-varying class prior is quite weak. It would be better to show simple empirical results why using time-varying priors can resolve the imbalance problem, and as a result it can produce the class-conditional functions. Second, the proposed algorithm is an extended version of LAS, and I think it is not a novel approach. Since the class incremental learning is highly related to the imbalance classification problem, it is straightforward to use logit adjustment to prevent the prediction bias.

2. In the large-scale dataset experiment, since all methods achieve poor results, it is hard to compare the results in proper way. In a practical side, the accuracy on large scale dataset should be high enough, but the proposed algorithm still achieves low average accuracy

**Questions:**

1. Why we should use class-conditional function in online CL? I think the motivation is not quite persuasive.

2. What is the main difference between the proposed method and LAS? I think it is just an application of LAS to class incremental learning with imbalance classification.